# ON THE COLLAPSE ERRORS INDUCED BY THE DETERMINISTIC SAMPLER FOR DIFFUSION MODELS

## ABSTRACT

Despite the widespread adoption of deterministic samplers in diffusion models (DMs), their potential limitations remain largely unexplored. In this paper, we identify *collapse errors*, a previously unrecognized phenomenon in ODE-based diffusion sampling, where the sampled data is overly concentrated in local data space. To quantify this effect, we introduce a novel metric and demonstrate that collapse errors occur across a variety of settings. When investigating its underlying causes, we observe a *see-saw effect*, where score learning in low noise regimes adversely impacts the one in high noise regimes. This misfitting in high noise regimes, coupled with the dynamics of deterministic samplers, ultimately causes collapse errors. Guided by these insights, we apply existing techniques from sampling, training, and architecture to empirically support our explanation of collapse errors. This work provides both intensive empirical evidence and theoretical analysis of collapse errors in ODE-based diffusion sampling, emphasizing the need for further research into the interplay between score learning and deterministic sampling, an overlooked yet fundamental aspect of diffusion models.

## 1 INTRODUCTION

Maximum likelihood-based generative modeling methods have demonstrated impressive capabilities for recovering data distributions, with diffusion methods being the latest advancement. A key advantage of diffusion models is their ability to achieve better diversity, whereas previous GAN-based methods (15) often struggle to fully capture the multi-modality of the data distribution (46; 35; 12). In diffusion models, the data distribution is learned by estimating the score function (the gradient of the log probability) through training denoisers. To enhance their performance, the score function is learned across various noise regimes and utilized in an annealing manner. The trained score models are then employed to sample from the data distribution, either via score-based Markov Chain Monte Carlo (MCMC) (58) or a reverse diffusion process (20; 59; 2). These models have achieved remarkable success in tasks such as super-resolution (29; 69; 50), text-to-image generation (46; 49; 43; 45), and video generation (21; 66; 6).

At first glance, diffusion models appear to be a complete solution: they possess strong theoretical foundations and achieve state-of-the-art practical performance, providing a double-layered validation. However, their practical behavior remains poorly understood. As researchers delve deeper into the success of diffusion models, they uncover intriguing phenomena—such as memorization (16; 65; 11; 8; 56), generalization (22; 30; 68), and hallucination (1; 25; 32)—that further complicate our understanding. A key observation is that, the Deep Neural Network (DNN) used for score learning plays a critical role in these phenomena. For instance, generalization in diffusion models has been attributed to certain inductive biases inherent in DNNs (22). Additionally, hallucination is often linked to underfitting in low noise regimes, where the target score function is complex (1). Conversely, memorization occurs when the model perfectly fits the optimal empirical denoiser (16), indicating an overfitting of the score function of the true population distribution.

Motivated by these insightful findings, we aim to continue this line of research by discovering and understanding the potential issues of the diffusion model paradigm. In particular, in addition to the previous works that mainly focus on the score learning, we also pay attention to the sampler in the diffusion model (i.e., the inference algorithm for DMs), especially the deterministic ones, e.g., DDIM (57), which have gained significant popularity due to their improved sampling efficiency,

controllable generation, and theoretical coherence. To systematically study this, we conducted extensive experiments by training hundreds of score-based DNNs on both real image and synthetic datasets, using different model sizes, dataset sizes, and sampling algorithms. Through our study, we identify a critical yet previously overlooked issue arising from the interplay between deterministic sampling and score learning. In specific, we study an overlooked phenomenon in diffusion models (see Fig. 1, 2, and Fig. 4 in (34)): Despite the success of deterministic samplers in diffusion models, **the samples they generate tend to become overly concentrated in certain regions of the data space**, compared to training samples and those generated by stochastic samplers (e.g., DDPM (20)). We term this phenomenon *collapse error*. We summarize the main findings of this paper as follows:

- **What is Collapse Error?** We study *collapse error*—an overlooked failure mode in deterministic diffusion samplers where generated samples become overly concentrated. Although hinted at in prior work (34), it has not been explicitly identified. To quantify this phenomenon, we introduce TID from first principles and use FID as a supplement, demonstrating the **universality** of collapse error across synthetic and real-world datasets under diverse settings.
- **Why Do They Occur?** We reveal that *collapse error* stems from the interplay between deterministic sampling dynamics and misfitting of the score function in high noise regimes. Through both extensive empirical experiments and theoretical analysis, show that this misfitting arises from the simultaneous learning of score functions in low and high noise regimes, a phenomenon we term the *see-saw effect*. To our knowledge, this is the **first** study to explicitly examine how score learning and sampler dynamics jointly contribute to an fine-grained failure mode in diffusion models.
- **How Is Our Explanation Validated?** To support our interpretation of collapse error, we evaluate several *existing* techniques originally proposed for other empirical purposes across three dimensions: sampling strategies, training methodologies, and model architectures. We find that these techniques coincide with our theoretical understanding and effectively reduce collapse error, thereby providing indirect validation of our hypothesis.

## 2 RELATED WORKS

**Mode Collapse.** When discussing collapse, it is natural to associate it with mode collapse in GANs (15), a well-known issue where the generator fails to capture the full multi-modality of the data distribution (9; 60; 72). Early work by Goodfellow et al. (15) introduced the GAN framework but acknowledged its instability during training, which often leads to mode collapse. Numerous efforts have sought to mitigate this issue. Techniques such as minibatch discrimination (51), unrolled GANs (40), and PacGAN (33) attempt to diversify the generator's output by improving the discriminator's ability to detect mode collapse. Wasserstein GANs (WGAN) (3) and its improved variant (WGAN-GP) (17) tackle mode collapse by reformulating the loss function to improve stability. Moreover, architectural innovations such as progressive growing of GANs (23) and BigGAN (7) have shown improved diversity in generated samples by leveraging better network designs. Despite these advances, mode collapse remains an active area of research, especially in tasks requiring high data complexity and diversity. While diffusion model collapse errors share some similarities with GAN mode collapse, a key distinction is that collapse errors in diffusion models can occur within a single mode. Furthermore, mode collapse in GANs is primarily caused by the discriminator dominating the training process (3; 17; 33), whereas the dynamics of collapse errors in diffusion models are fundamentally different.

**Samplers of Diffusion Models.** Stochastic samplers for diffusion models include annealing Langevin dynamics (58) and reverse stochastic differential equations (59). Empirical studies have shown that these stochastic methods suffer from high computational costs. In Langevin dynamics, a large number of steps is required to ensure adequate mode mixing (42; 4), while reverse stochastic differential equations demand high-resolution time step discretization for accurate sampling (59; 24). Fortunately, the reverse stochastic differential equation has an equivalent deterministic formulation (59; 57; 39), and experimental results demonstrate that deterministic samplers provide an improved sampling efficiency (53; 37; 55; 38; 73). A widely accepted explanation is that deterministic samplers produce straighter sampling trajectories, which facilitate the use of coarser time discretization (57; 44; 37; 53), and the ODE samplers are proven to be faster than SDE samplers theoretically (10). Numerous works have further extended deterministic samplers by designing better time discretization schemes, high-order solvers, and better diffusion processes (24; 13; 35; 34; 44).

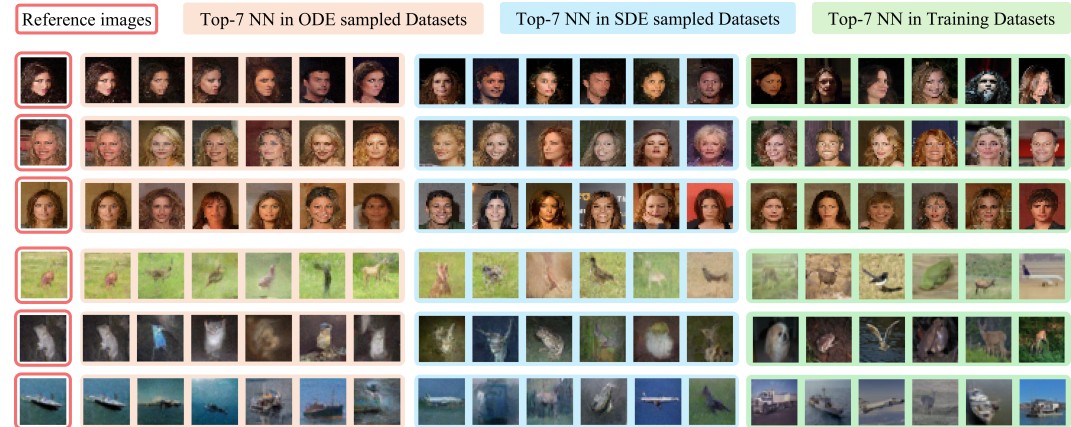

Figure 1: Collapse errors visualization at the sample level. The first column shows reference images from CIFAR10 and CelebA ODE-sampled datasets. The second, third, and fourth column show their top-7 nearest neighbors (NN) in ODE-sampled, SDE-sampled, and training datasets.

**Understanding & Explaining Diffusion Models through Score Learning.** Recent studies dissect diffusion behaviour almost exclusively through the lens of the score network itself. **Memorisation.** When the learned score *perfectly fits* the optimal empirical denoiser, the model can reproduce or leak training data, a risk documented for 2-D images (16; 65), 3-D medical volumes (11), and even exact sample extraction (8; 56). **Generalisation.** Conversely, *under-fitting* the empirical denoiser can improve test performance: geometry-adaptive harmonic bases (22), risk–bound analyses (30), and "fail-to-memorise" observations (68) link good generalisation to a deliberately imperfect score. **Hallucination.** *Extreme under-fitting* at low noise leads to hallucinated details, explained by mode interpolation (1) or structural artefacts in translation tasks (25). **Geometry.** Complementary work uncovers low-dimensional structure in the score field itself: subspace clustering (64), hidden Gaussian manifolds (31). **Score–Sampler Synergy.** While many perspectives attribute success or failure primarily to *score fitting*, another line of work observe that small misfitting in the high-noise regime can destabilize deterministic sampling (71), and SDE sampling mitigates score bias by injecting noise(67), though without fine-grained analysis. Our work advances this underexplored score–sampler interplay by uncovering a distinct and fine-grained failure mode, which we term *collapse error*. We trace collapse to a *see-saw effect* in diffusion training and the interaction between score approximation and deterministic sampling dynamics. Through extensive empirical evidence and theoretical analysis, we establish collapse error as a qualitatively new explanation of diffusion model failure.

## 3 BACKGROUND

### 3.1 DIFFUSION MODELS FOR GENERATIVE MODELING

Diffusion models define a forward diffusion process to perturb the data distribution $p_{data}$ to a Gaussian distribution. Formally, the diffusion process is an Itô SDE $d\boldsymbol{x}_t = \boldsymbol{f}(\boldsymbol{x}_t) + g(t)d\mathbf{w}$, where $d\mathbf{w}$ is the Brownian motion and $t$ flows forward from 0 to $T$. The solution of this diffusion process gives a transition distribution $p_t(\boldsymbol{x}_t|\boldsymbol{x}_0) = \mathcal{N}(\boldsymbol{x}_t|\alpha_t\boldsymbol{x}_0, \sigma_t^2\mathbf{I})$, where $\alpha_t = e^{\int_0^t f(s)ds}$ and $\sigma_t^2 = 1 - e^{-\int_0^t g(s)^2 ds}$. In the typical variance-preserving diffusion schedule, $\boldsymbol{f}$ and $g$ are designed such that $\lim_{t\to 0} p_t(\boldsymbol{x}) = p_{data}(\boldsymbol{x})$ and $\lim_{t\to T} p_t(\boldsymbol{x}) = \mathcal{N}(\boldsymbol{x}|\mathbf{0}, \boldsymbol{I})$. From this, it follows that $\lim_{t\to 0} \alpha_t = 1$, $\lim_{t\to 0} \sigma_t = 0$, $\lim_{t\to T} \alpha_t = 0$, and $\lim_{t\to T} \sigma_t = 1$. We refer to $t \to T$ and $t \to 0$ as high and low noise regimes, respectively. Diffusion models sample data by reversing this diffusion process, where $\nabla_{\boldsymbol{x}_t} \log p_t(\boldsymbol{x}_t)$ is required. To learn this term, a neural network $s_\theta$ is trained to minimize an empirical risk by marginalizing $\nabla_{\boldsymbol{x}_t} \log p_t(\boldsymbol{x}_t|\boldsymbol{x}_0)$, leading to the following loss:

$$L(\theta) = \mathbb{E}_{t\sim U(0,1), \epsilon\sim\mathcal{N}(\mathbf{0},\mathbf{I})} \sum_{n=1}^{N} \|s_\theta(\alpha_t\boldsymbol{x}_n + \sigma_t\epsilon, t) + \epsilon/\sigma_t\|^2.$$

To further balance the diffusion loss at different $t$'s, people usually adopt loss reweighing (24) or an alternate objective using $\epsilon$-prediction (20; 44), leading to the following well-known denoising score

matching (DSM) loss:

$$L(\theta, t) = \mathbb{E}_{\epsilon \sim \mathcal{N}(\mathbf{0}, \mathbf{I})} \sum_{n=1}^{N} \|s_\theta(\alpha_t \boldsymbol{x}_n + \sigma_t \epsilon, t) - \epsilon\|^2.$$

where $s_\theta(\cdot, t)$ can be viewed as the learned score function at time $t$. The DSM loss behaves differently between high and low noise regimes. In high noise regimes, since $\lim_{t \to T} \alpha_t = 0$, and $\lim_{t \to T} \sigma_t = 1$, the noisy observation of the data (i.e., $\alpha_t \boldsymbol{x}_n + \sigma_t \epsilon$) contains almost no data signals, and the $\epsilon$ can be easily inferred by nearly an identity mapping. In contrast, in low noise regimes, the noisy observation of the data is almost clean, making the $\epsilon$ prediction more challenging than the one in high noise regime.

## 3.2 SAMPLERS FOR DIFFUSION MODELS

To sample from the diffusion model, a typical approach is to apply a reverse-time SDE which reverses the diffusion process (2):

$$\mathrm{d}\boldsymbol{x}_t = [\boldsymbol{f}(\boldsymbol{x}_t) - g(t)^2 \nabla_{\boldsymbol{x}_t} \log p_t(\boldsymbol{x}_t)]\mathrm{d}t + \mathrm{d}\bar{\mathbf{w}},$$

where $\mathrm{d}\bar{\mathbf{w}}$ is the Brownian motion and $t$ flows forward from $T$ to 0. For all reverse-time SDE, there exists corresponding deterministic processes which share the same density evolution, i.e., $\{p_t(x_t)\}_{t=0}^{T}$ (59). In specific, this deterministic process follows an ODE:

$$\mathrm{d}\boldsymbol{x}_t = [\boldsymbol{f}(\boldsymbol{x}_t) - \frac{1}{2}g(t)^2 \nabla_{\boldsymbol{x}_t} \log p_t(\boldsymbol{x}_t)]\mathrm{d}t,$$

where $t$ flows backwards from $T$ to 0. The deterministic process defines a velocity field, $v(\boldsymbol{x}, t) = [\boldsymbol{f}(\boldsymbol{x}_t) - \frac{1}{2}g(t)^2 \nabla_{\boldsymbol{x}_t} \log p_t(\boldsymbol{x}_t)]$. Here, we also define the velocity field predicted by score neural network, $s_\theta$: $v_\theta(\boldsymbol{x}_t, t) = \boldsymbol{f}(\boldsymbol{x}_t) - \frac{1}{2}g(t)^2 s_\theta(\boldsymbol{x}_t, t)$.

ODE-based deterministic samplers offer distinct advantages over stochastic methods. First, they achieve efficient sampling with drastically fewer steps compared to stochastic samplers, while maintaining high-quality outputs (59; 57). Besides, their deterministic nature ensures reproducible results from fixed initial noise, crucial for controlled generation tasks such as latent space interpolation (57; 50). Moreover, deterministic trajectories mitigate error accumulation in low-step regimes, delivering more stable sample fidelity than stochastic counterparts (24). Additionally, the ODE framework provides theoretical coherence, enabling rigorous stability analysis and integration with advanced solvers (37; 38) for accelerated sampling.

## 4 INTRODUCTION OF COLLAPSE ERRORS IN DIFFUSION MODELS

In this section, we introduce collapse errors both conceptually and visually, explaining how they can be observed at both the individual sample level and the distributional level.

**Collapse Errors at the Sample Level.** To illustrate the collapse phenomenon intuitively, we visualize several collapsed samples when we train typical diffusion models on CIFAR10 and CelebA datasets. The training of diffusion models follows the typical Variance-Preserving $\epsilon$-prediction score matching schedule (59), with the only modification being the dataset size, which is set to 20,000. Detailed experimental settings can be found in Appendix.B.1. In Fig. 1, we show three collapsed samples from each dataset, along with their top-7 Nearest Neighbors (NN) search by $l_2$ norm, in the ODE sampled, SDE sampled, and training datasets. We observe that, although ODE-sampled images exhibit good quality, their nearest neighbors tend to be exhibit similar attributes than those from SDE-sampled and training datasets. For example, ODE samples share nearly identical facial orientation, whereas stochastic samplers yield variation. Besides, ODE samples display uniform hair color, gender, and expression, whereas others do not. While these differences are hard to capture

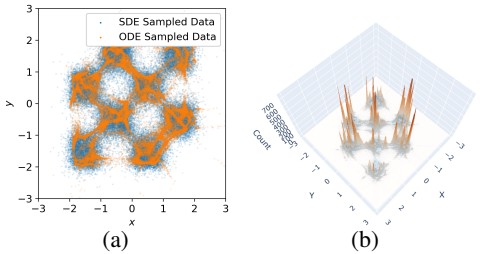

Figure 2: ODE and SDE sampled data in scatter plots (left) and histograms (right).

quantitatively through visuals alone, they point to *conceptual demonstration of the practical impact* of collapse error. To quantify this, we will design specific metric from the first principle. More collapsed samples can be found in Appendix. B.2.

**Collapse Errors at the Distribution Level.** Now that we have discussed the collapse errors at the sample level, we extend the concept of collapse errors to the distribution level. Specifically, we visualize the collapse errors in lower-dimensional data. Fig. 2 shows an example of collapse error when training a MLP on a 2D chessboard-shape distribution. Detailed experimental settings can be found in Appendix.C.1. We observe that, compared to SDE sampled data points, the ODE sampled data points are more concentrated in certain regions. Specifically, in Fig. 2a, we observe the clustering of ODE sampled data point cannot cover the SDE sampled data points, indicating the ODE sampled data points exhibit less divergence. We emphasize that *such phenomenon can be observed from existing work*. For example, in Fig. 4 of (34), deterministic samplers yield non-uniform samples on the same chess-board dataset. When we look at the histogram of this 2D clustering plot, shown in Fig. 2b, we find that the ODE sampled distribution exhibits sharp peaks in specific regions, in contrast to the SDE-sampled distribution, which indicates the samples are concentrated in those regions. Collapse errors on more synthetic datasets can be found in Appendix. C.2.

# 5 THE CAUSES OF COLLAPSE ERRORS

We first introduce a quantitative metric to evaluate collapse errors and explore key influential factors. We then present intensive empirical evidence to identify the root causes behind collapse errors.

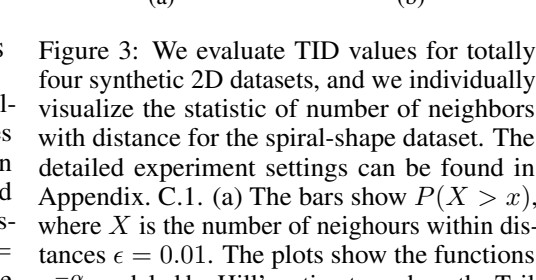

(a)  (b)

Figure 3: We evaluate TID values for totally four synthetic 2D datasets, and we individually visualize the statistic of number of neighbors with distance for the spiral-shape dataset. The detailed experiment settings can be found in Appendix. C.1. (a) The bars show $P(X > x)$, where $X$ is the number of neighours within distances $\epsilon = 0.01$. The plots show the functions $x^{-\alpha}$ modeled by Hill's estimator, where the Tail Index $\alpha$ quantifies the heaviness of tail. (b) The Tail Index Difference ($TID(\epsilon)$) measured on various datasets. A higher TID value at specific $\epsilon$ distances correspond to more severe collapse.

## 5.1 QUANTIFICATION OF COLLAPSE ERRORS

In the previous Sec. 4, we showed that when collapse errors occur, we can identify some samples whose neighbors are more similar to them. In other words, by fixing a distance, we can find samples that have more neighbors within this distance. Formally, let the training dataset be $D = \{x_1, x_2, ..., x_N\}$, where $x_i$ is a data and $N$ is the dataset size, we define the number of neighbors of $x_i$ within a distance $\epsilon$ as:

$$n_i(D, \epsilon) = \#\{x_j \mid d(x_i, x_j) \leq \epsilon, j \in [1, N]\},$$

where $d(\cdot, \cdot)$ is a distance metric. In this paper, we specifically use the $l_2$ norm as the distance metric throughout, as it is more essential for capturing collapse errors in diffusion models. We further discuss the choice of distance metrics in Appendix. A and I.

In Fig. 3a, we observe that when collapse occurs, the ODE-sampled dataset tends to have more neighbors within a short distance than the training dataset. This leads to a heavier tail in the survival function $P(\# \text{ of neighbors} > x)$, reflecting stronger local concentration. To rigorously measure this effect, we employ Hill's estimator (19):

$$\alpha(D, \epsilon) = \frac{1}{N} \sum_{i=1}^{N} \log \frac{\hat{n}_k(D, \epsilon)}{\hat{n}_N(D, \epsilon)},$$

where $\hat{n}_k$ denotes the $k$-th largest statistic of $n$ after sorting in descending order. Intuitively, a smaller $\alpha$ corresponds to a heavier tail, i.e., stronger concentration of neighbors. To compare the relative heaviness between two datasets, we define the Tail Index Difference (TID): $TID(D_{SD}, D_{TD}, \epsilon) = \alpha(D_{TD}, \epsilon) - \alpha(D_{SD}, \epsilon)$, where $D_{TD}$ and $D_{SD}$ is the training datasets and sampled dataset, respectively. A larger TID indicates that the sampled data are more locally concentrated than the training data. For clarity of presentation, we often omit the inside $D_{TD}, D_{SD}$ and $\epsilon$, as these will be understood from the context.

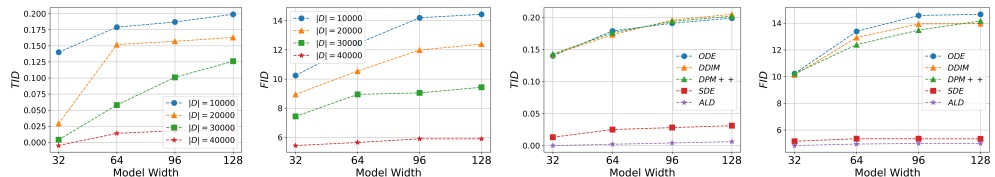

Figure 4: TIDs/FIDs evaluated on ODE sampled images generated by diffusion models trained on CIFAR10 dataset across various training settings, containing model width and samplers.

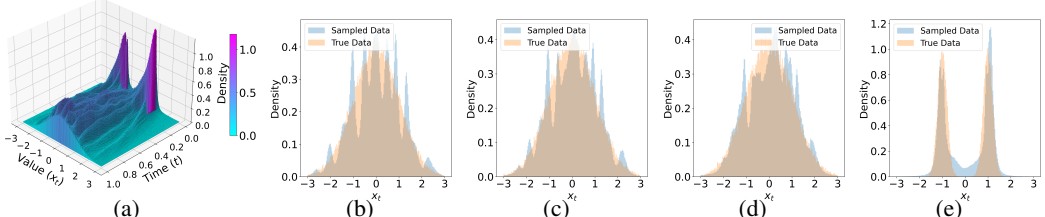

Figure 5: In the high-dimension MoG dataset setting, we visualize the density evolution of the first dimension of the data during an ODE sampling. (a) The evolution of the probability density across timesteps, starting from the Gaussian prior to the final target distribution. (b-e) The marginal distribution $x_t$ between sampled data (blue) and ground truth training data (orange) at specific timesteps $t = 0.8, 0.6, 0.4, 0.0$.

Fig. 3b shows the TID plot along $\epsilon$ across various synthetic datasets. The detailed description and visualization of these synthetic datasets, and experimental settings can be found in Appendix. C.1. We found that in certain interval of $\epsilon$, the TID is above 0, showing that the ODE sampled dataset suffers from a collapse error. While TID involves pairwise distance computation, it can be **efficiently** estimated using a subset of the dataset, due to the *scale-invariant* property of Hill's estimator, making the TID metric practical for large dataset. Importantly, while TID is derived from first principles to directly quantify collapse error, we also find its trends to be consistent with FID measurements, further validating the reliability of TID. Therefore, we report FID together with TID on image data. We put more discussion of relation between TID and FID in Appendix. A and L.

## 5.2 INFLUENTIAL FACTORS OF COLLAPSE ERRORS

We conduct intensive experiments on real image datasets and use TID and FID as metrics to evaluate the collapse error. We find that colllapse error occur in a wide range of training settings. Fig. 4 shows the tendency of TID/FID along different training settings, including model width, different samplers, and dataset size. Our experiments follow the standard score-matching training in (59). We give the details of these experiments in Appendix. B.1. In this subsection, we mainly demonstrate our findings on collapse trends along these training factors on CIFAR10. We also provide experimental results on CelebA in Appendix. D, which demonstrate similar TID trends. Especially, we observe that the choice of sampler plays a critical role in collapse errors, as measured by TID/FID. Deterministic samplers such as ODE (59) and DDIM (57) consistently exhibit higher TID/FID values, suggesting increased sample concentration. In contrast, stochastic samplers like SDE (59) and ALD (58) maintain low TID/FID values across settings, preserving diversity.

## 5.3 COLLAPSE ERRORS PROPAGATES DURING SAMPLING

Before discussing the misfitting in diffusion models, we demonstrate how collapse errors occur in a specific case. To better illustrate and analyze this phenomenon, we conduct experiments on a high-dimension Mixture of Gaussian setting, which provides an analytical form of score function for further analysis. We put the derivation of score function for our MoG setting in Appendix. E.2. Specifically, we suppose a synthetic $n$-dimension MoG dataset by:

$$x_0 \sim 0.5 \times \mathcal{N}(x_0| -1_n, 0.2I_n) + 0.5 \times \mathcal{N}(x_0|1_n, 0.2I_n),$$

where $1_n$ represent a vector filled with ones with a length of $n$ and $I_n$ is an identity matrix with a size of $n \times n$. The details of the experimental settings can be found in Appendix. E.1. While this section focuses on a specific data distribution to better illustrate how collapse errors occur, we provide additional experimental results in Appendix.F with similar phenomena across other settings.

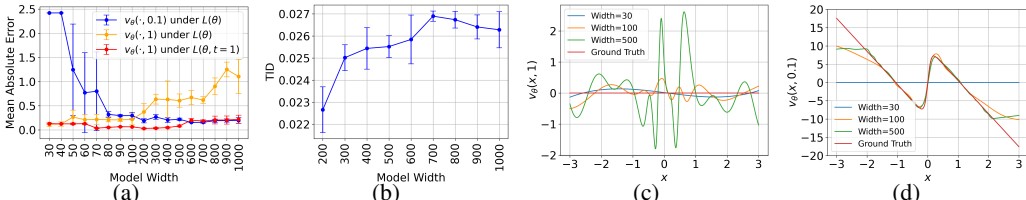

Figure 7: Evaluations of diffusion models trained on high-dimensional MoG when the models are MLPs with increasing widths. (a) Mean absolute errors of $v_\theta(\cdot, 1)$ and $v_\theta(\cdot, 0.1)$ accross different MLP widths trained under $L(\theta)$ and $L(\theta, t = 1)$. (b) $TD(\epsilon = 0.02)$ evaluated on ODE sampled datasets with varying model width. (c) and (d) visualize the learned $v_\theta(\cdot, 1)$ and $v_\theta(\cdot, 0.1)$, respectively, across different MLP widths.

**Collapse Errors Occur at the Beginning of Sampling.** In Section 4, we characterized collapse errors as sharp-peak patterns at the distribution level. In this experiment, we find that collapse errors occur early in the ODE sampling process. Fig. 5b shows the intermediate sampled distribution from $t = 1$ to $t = 0.8$. We find that even in such early stages of sampling, the collapse errors are already significant, as indicated by the sharp peaks in the distribution. In addition, as shown in Fig. 5a, the collapse errors begin as soon as the the ODE sampling starts, marked by the presence of sharp peaks in the density. Similar early-onset patterns have also been observed in prior work (Fig. 4 of (34)).

**Collapse Errors Propagate along $t$.** During the ODE sampling, collapse errors not only occur at the begin of sampling but also propagate and intensify as the sampling progresses. For example, in Fig. 5a, we observe the formation of sharp peaks in $x_t$ that propagate along $t$, creating distinct ridges in the 3D visualization. These ridges represent regions where the probability density becomes highly concentrated as sampling progresses.Such phenomenon can be observed from existing work. For example, Fig. 4 of (34) demonstrates that sample concentration occurs early in the sampling process particularly in score matching, which strongly aligns with our observations.

**Velocity Error Propagtes along determinstic sampling steps $x_t$.** To understand why collapse errors tend to accumulate during sampling, we examine the velocity field predicted by the diffusion model,

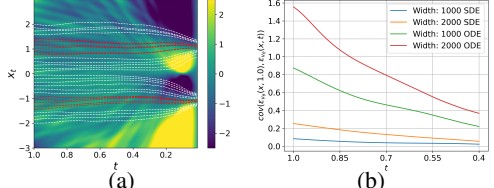

Figure 6: (a) Visualization of first dimension of velocity field ($v_\theta(\boldsymbol{x}, t)[: 1]$) when the target distribution is a high-dimension MoG. Here, the other dimension of $x_t$ ($x_t[1 :]$) are fixed by a standard Gaussian noise and the velocity is calculated along the first dimension of $x_t$ ($x_t[: 1]$). (b) Velocity error covariance across sampling step $\boldsymbol{x}_t$. The covariance is calculated by comparing the error vectors of $v_\theta(\boldsymbol{x}_t, t)$ and $v_\theta(\boldsymbol{x}_{1.0}, 1.0)$. The tested points $\boldsymbol{x}_1$ are sampled from high-dimension standard Gaussian.

$v_\theta(x_t, t)$, as shown in Fig. 6a. At $t \approx 1$, the velocity field displays oscillatory misfitting along the $x_t$-axis, but appears relatively static across $t$, suggesting that erroneous local patterns in velocity do not correct over time. As a result, sampled data trajectories converge and collapse into narrow paths, as visualized by the red dashed lines. To quantify how these errors persist, we compute the covariance of velocity prediction errors between different time steps, shown in Fig. 6b. We observe that ODE-based samplers exhibit significantly higher error covariance across $t$, indicating that errors made at earlier steps propagate and accumulate. In contrast, SDE-based samplers exhibit minimal covariance, suggesting their inherent stochasticity helps decorrelate errors over time, thereby mitigating collapse. Additional results across datasets and architectures supporting this finding are provided in Appendix F.

In summary, we show that the collapse errors occur in the early sampling stage and are retained as the sampling progresses. We also identify that these early-stage collapse errors are caused by the misfitting of the predicted velocity field, which will be investigated in the next subsection.

## 5.4 MISFITTING IN HIGH NOISE REGIMES

In Sec. 3, we showed that the training dynamics of diffusion models vary across $t$. In high noise regimes (i.e., $\alpha_t \to 0$ and $\sigma_t \to 1$), the diffusion model predicts noise from data that is nearly Gaussian noise. Specifically, under the $\epsilon$-prediction training objective, the task of diffusion models in high noise regimes is a trivial identity mapping. At first glance, one might expect that the simplicity of

the training objective in large noise regimes would lead to perfect learning. However, our experiments in the previous subsection reveal that diffusion models can misfit even this straightforward target function, as illustrated by the oscillatory pattern of $v_\theta(x_t, 1)$ in Fig. 6a, and the large error covariance shown in Fig. 6b. This counterintuitive finding motivates us to investigate the underlying causes of misfitting in high noise regimes.

We conduct experiments on both synthetic datasets and real image datasets. For synthetic datasets, we use a high-dimensional MoG as the training dataset to ensure an analytical score function, and use a 2-layer MLP as the score prediction neural network (details in Appendix. E.1). In Fig. 7a, we show the Mean Absolute Errors (MAE) of $v_\theta(\cdot, 1)$ (High Noise Regime), and $v_\theta(\cdot, 0.1)$ (Low Noise Regime), when we increase the MLP width. We observe a counterintuitive trends in high noise regimes: as the model width increases, the MAE of $v_\theta(\cdot, 1)$ increases, despite the task being a trivial identity mapping. One may consider that the errors in high noise regime are due to the larger model capacity. However, when we train the diffusion models only in high noise regimes (under $L(\theta, t = 1)$) with growing model capacities, we do not observe significant errors, as marked as the red line in Fig. 7a. We refer to this phenomenon as a **see-saw effect** in diffusion model training: In training a diffusion model in both low and high noise regime, the learning in low noise regimes can adversely affect the learning in high noise regimes. We also confirm the misfitting in high noise regime by visualizing $v_\theta(\cdot, 1)$, shown in Fig. 7c. To validate the universality of the see-saw phenomenon, we provide theoretical results in Proposition 1 and additional experimental results in Appendix. G.3.

We also find similar trends on CIFAR10, as shown in Fig. 8. It is noteworthy that unlike the synthetic MoG dataset, the score function for real datasets is unknown, so we cannot directly calculate the velocity errors. Instead, we use DSM loss which serves as the MSE from the optimal empirical denoiser (63). In specific, we evaluate the predicted score function in high and low noise regimes by $L(\theta, t = 1)$ and $L(\theta, t = 0.1)$. We give detailed training settings in Appendix. G.1. As shown in Fig. 8a, in low noise regimes, larger models achieve lower DSM loss, as expected. However, in the high

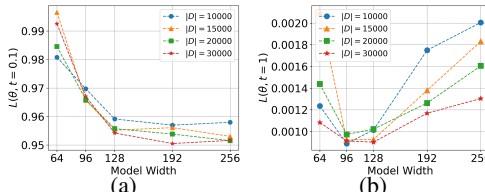

(a)  (b)

Figure 8: Diffusion loss $L(\theta, t = 0.1)$ (left) and $L(\theta, t = 1)$ (right) when the diffusion models are trained on CIFAR10 with various settings on model widths and dataset sizes.

noise regime, the DSM loss increases with model size, indicating misfitting. While increasing the dataset size reduces the DSM loss in both regimes, larger models still exhibit slightly higher errors in the high noise regime even with larger datasets. We also conduct the same experiment on CelebA and find similar see-saw phenomenon on DSM loss, and we put them in Appendix. G.

# 6 VALIDATING THE CAUSE OF COLLAPSE VIA EXISTING METHODS

In the previous section, we demonstrate that the collapse errors arise from two main factors: (1) The error in the velocity field tends to propagate along $t$, and (2) when the model becomes capable of learning the complex score function in low noise regimes, the errors in high noise regimes tend to increase. Building on these insights, in this section, we introduce several *existing* techniques to mitigate collapses errors from three perspectives, including sampling, training and model architecture. We conduct experiments on a high-dimensional MoG dataset and use TID to evaluate the effect of the introduced techniques on mitigating collapse errors (details in Appendix. H.1). Experiments on

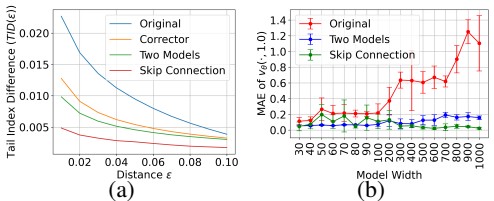

(a)  (b)

Figure 9: (a) TID values evaluated across different $\epsilon$ comparing the techniques and the original method, when the MLP width is 1000. (b) MAE of $v_\theta(\cdot, 1.0)$ trained by MLPs with different widths comparing original method, two-model training, and skip connection.

CIFAR10 and CelebA are put in Appendix H.2. *It is important to note that our goal is not to propose these techniques as definitive solutions to completely eliminate collapse errors but rather to provide supporting evidence that reinforces our understanding of their underlying causes.*

**Sampling Techniques.** We have shown that velocity errors tend to propagate along $t$, causing the data point to be influenced by similar errors, which bias its sampling trajectory and ultimately lead to collapse errors in the final generated data. Building on this insight, we are motivated to introduce stochasticity during sampling, which allows the sampled data points be influenced by random errors during sampling. We find that the predictor-corrector sampler is effective in address collapse errors. In a predictor-corrector sampler, the ODE sampler is combined with an extra score-based MCMC stage:

$$\boldsymbol{x}_t^{m+1} = \boldsymbol{x}_t^m + \epsilon_t^m s_\theta(\boldsymbol{x}_t^m, t) + \sqrt{2\epsilon}\boldsymbol{z}_t^m,$$

where $\boldsymbol{z}_t^m$ is a random standard Gaussian noise, $\epsilon_t$ is the step size of the MCMC, and $m = 1, 2, ..., M$. This iteration can be repeat multiple times to improve the accuracy of MCMC. We follow the official implementation in (58), where the step size $\epsilon_t^m$ is set to be $\alpha_t(r\|z\|_2/\|s_\theta(\boldsymbol{x}_t, t)\|_2)^2$, where $r$ is a predefined signal-to-noise ratio, and $M$ is set to be 1.

**Training Techniques.** We have shown that despite the trivial target function in high noise regimes, the model tends to misfit it when it is capable of fitting complex function in low noise regimes. We hypothesize that training in low noise regimes adversely affect that in high noise regimes. To support this hypothesis, we propose a technique to separate the training for high and low noise regimes. In specific, without modifying the model architecture, we training the original model under smaller $t$, and a duplicate model under larger $t$. In specific, the training objective is:

$$L'(\theta_1, \theta_2) = \mathbb{E}_{t\sim U(0,t')}L(\theta_1, t) + \mathbb{E}_{t\sim U(t',1)}L(\theta_2, t),$$

where $\theta_1$ and $\theta_2$ are the parameters of the two models, and $t'$ is a value within $(0, 1)$, indicating the separation point between the high and low noise regimes.

**Model Architecture.** We have shown that the target function in high noise regimes is nearly an identity mapping in the $\epsilon$-prediction objective. This motivates us to incorporate skip connection into model architecture (24), since skip connection can provide an identity mapping as a precondition. In specific, we propose the model architecture with skip connections by:

$$\hat{s}_\theta(\boldsymbol{x}_t, t) = c_{\theta_1}^1(t)\boldsymbol{x}_t + c_{\theta_2}^2(t)s_\theta(\boldsymbol{x}_t, t),$$

where $c_{\theta_1}^1(t)$ and $c_{\theta_2}^2(t)$ are learnable MLPs with parameters of $\theta_1$ and $\theta_2$, respectively, and $s_\theta(x_t, t)$ is a neural network. Fig. 9 shows the effectiveness of these techniques on mitigating collapse errors. By applying techniques on two-model training and skip connections, the diffusion model no longer misfit in high noise regimes when model size increases.

# 7 CONCLUSION AND LIMITATION

In this paper, we introduced *collapse error*, a previously unexplored error pattern in diffusion models where deterministic samplers overly concentrate probability mass in data space. Although hints of this phenomenon appear in prior work, its mechanism and consequences have not been systematically studied. To quantify collapse, we proposed the Tail Index Difference (TID), derived from first principles, and further employed FID as a complementary metric, since we find FID can also capture collapse indirectly (Appendix. A). Thanks to the scale-invariant property of Hill's estimator, TID can be computed efficiently, making it applicable to large datasets. We demonstrated the practical impact of collapse error through both quantitative and qualitative evidence: numerically, collapse degrades traditional metrics such as FID/TID; conceptually, collapsed samples exhibit excessive semantic similarity. More fundamentally, collapse reflects *an intrinsic flaw* in distribution modeling: the learned distribution becomes spuriously concentrated, deviating from the true target. We attributed collapse to the interplay between deterministic samplers and score misfitting, and we theoretically showed a see-saw effect in score matching training that induces such misfitting. Finally, guided by our first-principle understanding, we proposed mitigation techniques that proved effective in alleviating collapse, thereby validating our reasoning.

**Limitations.** Our work does not propose a novel training algorithm or sampler specifically designed to eliminate collapse error. While we conducted extensive controlled experiments and validated our observations against prior findings, our study does not exhaustively cover collapse across large-scale datasets, broader model sizes, or extended training regimes. Such a systematic exploration would require scaling experiments far beyond the scope of this work. We instead focus on controlled settings to isolate and rigorously analyze collapse, leaving large-scale investigations for future work.

## 8 REPRODUCIBILITY STATEMENT

Source code necessary to reproduce our experiments is provided in the supplementary material submitted with this paper. Detailed descriptions of datasets, preprocessing steps, model configurations, and training procedures are included in Appendix B.1, C.1, E.1, G.1, H.1, ensuring full reproducibility of our results.

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

# A  DISCUSSION

**Score Learning & Deterministic Samplers**  In this study, we identify and investigate collapse errors in diffusion models when using deterministic samplers. We find that collapse errors arise from the interplay between deterministic sampling dynamics and overfitting of the score function in high noise regimes. While both factors contribute to collapse errors, our analysis suggests that score learning plays a more significant role, whereas deterministic samplers primarily act as a trigger. In our preliminary experiments, when we apply different deterministic samplers to the same learned score model, we observe no significant differences in collapse severity. We put the preliminary results in Appendix K. Although the variations across deterministic samplers are subtle, further investigation into their impact on collapse errors remains an interesting direction for future research.

**Distance Metric**  To evaluate collapse errors, we calculate distances among samples using the $l_2$ norm in the data space, whereas many existing works measure Fréchet distances (14) in the feature space. Our motivation for directly applying the $l_2$ norm is that the reverse diffusion process operates in the data space, making it the natural setting where collapse errors occur. Nevertheless, it remains an interesting direction to investigate collapse errors in the feature space. Our preliminary experiments reveal that when collapse errors occur, their statistics in the feature space become biased, leading to mean shifts. We put the preliminary results in Appendix. I. We hypothesize that in the feature space, each channel represents local patterns in the images; thus, when collapse errors occur, the presence of certain local patterns increases or decreases, resulting in feature bias. As a result, standard metrics such as FID and Inception Score also respond to collapse, indirectly capturing its effect. Consequently, addressing collapse errors may lead to improved performance as reflected by these metrics. We leave the collapse errors in feature space as an open question for further research.

**Time Embedding**  To address error propagation along $t$, our initial approach involved using time embedding methods; however, we found them to be less effective. Our experiments, shown in Appendix. F, revealed that velocity errors can still propagate over short periods of $t$ even using time embedding, leading to collapse errors. From another perspective, DNNs actually generalize through interpolation (41; 5; 70). If our goal is to eliminate error propagation, it implies preventing the DNN from interpolating along $t$. However, this raises an important question: would limiting interpolation along $t$ affects diffusion models generalization? We leave this as an open question for further research.

# B  EXPERIMENTS ON REAL IMAGE DATASET

## B.1  EXPERIMENTAL SETTINGS

In this section, we introduce our experimental settings on real image dataset, containing CIFAR10 (27), CelebA (36), and MNIST (28).

**Diffusion Process**  In this paper, we follow the typical variance-preserving diffusion process predefined in (59). This diffusion process serves as the default setting for all experiments, unless otherwise specified. In specific, the diffusion process is defined by the following stochastic differential equation:

$$\mathrm{d}\mathbf{x} = -\frac{1}{2}\beta(t)\mathbf{x}\,\mathrm{d}t + \sqrt{\beta(t)}\,\mathrm{d}w,$$

where $t$ progresses from 0 to 1, $\mathbf{x}$ represents the data vector at time $t$, and $\mathrm{d}w$ denotes the brownian motion. The time-dependent noise variance function $\beta(t)$ controls the amount of noise added to the data over time, and it is defined as:

$$\beta(t) = \bar{\beta}_{\min} + t\left(\bar{\beta}_{\max} - \bar{\beta}_{\min}\right),$$

where $\bar{\beta}_{\min} = 0.1$ and $\bar{\beta}_{\max} = 20$. It is important to note that in this paper, we primarily focus on the typical VP diffusion process to maintain variable control. However, this does not imply that collapse errors occur only in the VP diffusion process. Appendix J presents our preliminary results of collapse errors on other diffusion processes.

**Model Architecture**    For CIFAR10 and CelebA, we adopt the a U-Net architecture (47). We set the default setting of U-Net as the one introduced in (59), with a modification to the channel multiplier, changing it from the default setting of (1,2,2,2) to (1,2,2). This adjustment is made to accommodate experiments on lower resolutions. Retaining the original three-layer U-Net configuration would constrain our resolution choices to multiples of 8. By transitioning to a two-layer U-Net, we enable the use of resolutions that are multiples of 4, which facilitates experimentation with lower data dimensions. Additionally, we adjust the model size by tuning the config.model.nf parameter, which controls the model width. For MNIST, the model was a four-layer U-Net, consisting of an encoder and decoder. The encoder included four convolutional layers (kernel size 3, LogSigmoid activation) and a MaxPooling layer ($2 \times 2$) for downsampling. For channel sizes for each convolutions are 32, 64, 128, 256, from the top level to the bottom level. The decoder mirrored this structure with transpose convolutions for upsampling and used skip connections to combine features from the encoder. A final output convolution layers layer to reconstruct the image. To feed the time variable, the time variable $t$ was expanded and concatenated to the input $x$ as an additional channel.

**Dataset Construction**    To illustrate the collapse error, we reduce the original CIFAR10 and CelebA datasets to a size of 20,000 samples. To better observe the collapse phenomenon, no training techniques such as data augmentation are applied. Additionally, training datasets with varying image sizes are generated by down-sampling the original data. For CIFAR10 dataset, we use the defaut settings in pytorch.

**Training**    We set the epoch number to be 48,000 for CelebA and CIFAR10, and 1200 for MNIST. We use Adam optimizer (26) with a learning rate of 5e-3 for CIFAR10 and CelebA, and 2e-4 for MINST. We use stochastic gradient decent with a batch size of 128 for CIFAR10 and CeleA, and 60 for MNIST. The training objective is the $\epsilon$-prediction objective as discussed in Sec. 3. All experiments were run on 8 NVIDIA RTX4090 GPUs.

**Sampling and Evaluation**    To sample from the trained diffusion model, we use ODE samplers with 100 steps and SDE samplers with 1,000 steps using Euler's method, where $t$ is discretized evenly within (0,1). We adpot the same ODE samplers and SDE samplers from (59). We further discuss the samplers in Appendix. K. We apply these samplers for all experiments unless otherwise specified. To evaluate $TID(D_{TD}, D_{SD}, \epsilon)$, we obtain $D_{TD}$ by randomly sample 2,000 data from the training dataset, and we obtain $D_{SD}$ by sampling 2,000 data from the diffusion model. We set the default distance $\epsilon$ parameter in TID for CIFAR10 and MNIST by 0.2, and for CelebA by 0.25.

When we conduct experiments on TID trends, we fix a default training settings and do ablation study on other dimension. The default setting is "image size=16, Model Width=128, Dataset size=20,000, training epoch=48,000".

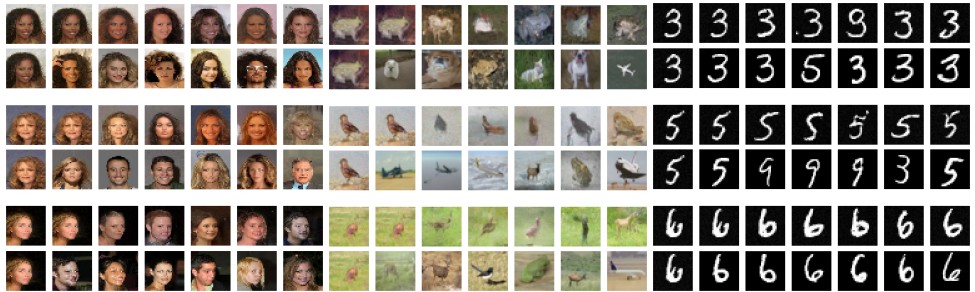

Figure 10: The first row shows reference images sampled from the diffusion model (using ODE sampling) and their nearest neighbors in the generated dataset. The second row shows the reference images and their nearest neighbors in the training dataset. For left to right, each column represents the dataset of CelebA, CIFAR10, and MNIST.

## B.2    More Collapse Samples in Real Image Dataset

In Fig. 10, we show additional collapse samples to supplement Fig. 1. It can be observed that the nearest neighbors in the generated dataset (first row) are much more similar to the sampled images

compared to the nearest neighbors in the training dataset (second row). This highlights a collapse phenomenon, where ODE sampled images are overly concentrated in certain regions of the data space.

# C  EXPERIMENTS ON 2D SYNTHETIC DATASET

## C.1  EXPERIMENTAL SETTINGS

In this section, we introduce our experimental settings on synthetic datasets.

**Dataset Construction**   The chessboard-shaped dataset is generated to form a $4 \times 4$ grid pattern, mimicking a chessboard, where data points are concentrated in alternate cells. Each cell is $1 \times 1$ unit in size. The points within each cell are uniformly distributed.The spiral-shaped dataset consists of points along a single spiral curve. The spiral is generated by varying the radius and the angle of each point. The radius increases linearly from 0 to 2 units as the angle progresses from 0 to $4\pi$ (representing two full turns). Then Gaussian noise of standard deviation 0.1 added to their coordinates. The semi-circle-shaped dataset comprises two semi-circles with radius of 1, positioned at slightly different vertical offsets. The first semi-circle is centered at $(0.5, 0.1)$, and the second one is centered at $(-0.5, -0.1)$. Points are evenly distributed along these arcs, with Gaussian noise of standard deviation 0.1 added to each coordinate. The 2D Mixture of Gaussians (MoG) dataset is generated by defining 6 Gaussian components, each with a mean placed in a circular pattern with a radius of 2 and a standard deviation of 0.2. The weights of each component are set equally, and the covariance matrices are diagonal with the same variance for both dimensions. All the dataset sizes are set as 500,000.

**Model Architecture**   We use a three-layer Multilayer Perceptron (MLP) with 100 neurons in each layer and Tanh activation functions. The time variable $t$ is concatenated to the data input $x$.

**Training**   The model was trained in stochastic gradient decent with a batch size of 2000 and using the Adam optimizer (26) with a learning rate of $5 \times 10^{-3}$, and the training was conducted over 10,000 iterations.

## C.2  COLLAPSE ERRORS ON 2D SYNTHETIC DATASET

In Fig. 11, we visualize collapse errors at distribution level on more 2D synthetic datasets to supplement Fig. 2

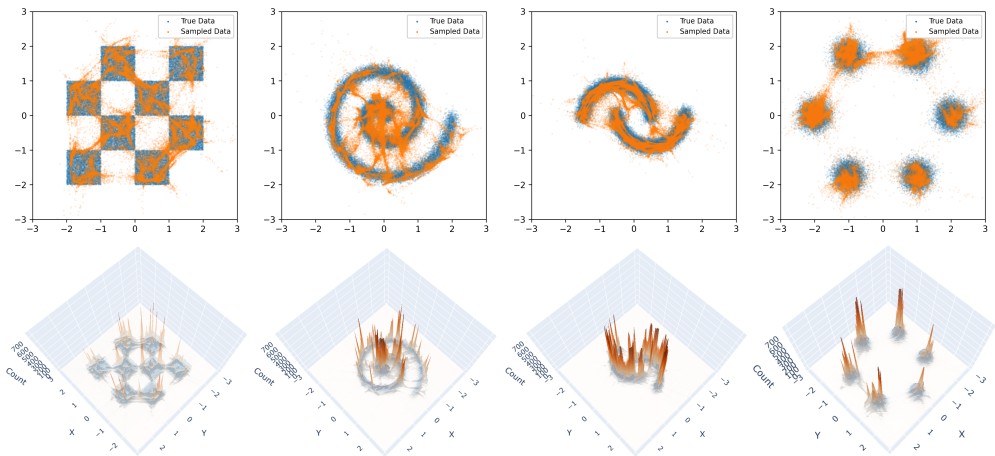

Figure 11: Comparison of ODE sampled data and true data in both scatter plots (top row) and histograms (bottom row). From left to right, the true data distribution are chessboard, spiral, semi-cicle and MoG-shaped distribution. Blue represents the sampled data, and orange represents the true data. The collapse phenomenon can be observed as the sampled data concentrates in specific regions, leading to a sharp peak in the histogram.

## D  TID ON CELEBA

To supplement Fig. 4, we evaluate the TID on CelebA on various experimental settings, as shown in Fig. 12. The details of experimental settings can be found in Appendix. B.1.

We observe that the TID values evaluated on almost all training settings are larger than 0, indicating the university of collapse errors. Observations from Fig. 4 and 12 are summarized as follows:

**Model Width**  TID values increase as the model width grows. This trend is particularly significant for smaller dataset sizes (e.g., 10,000, 20,000, 30,000), where the collapse error becomes more severe with larger model capacities. For larger dataset sizes (e.g., 40,000), the growth in TID values is less evident, suggesting that larger datasets mitigate collapse errors to some extent, even when model width increases.

**Dataset size**  TID values decrease as dataset size increases. For all model widths, smaller datasets (e.g., 10,000 samples) exhibit significantly higher TID values, indicating more pronounced collapse errors. In contrast, larger datasets (e.g., 40,000 samples) result in near-zero TID values, highlighting the importance of dataset size in reducing collapse errors.

**Dimension**  TID values grow with higher data dimensions. For smaller dataset sizes (e.g., 10,000), the TID values increase steeply as data dimensions increase, indicating that collapse errors are amplified in higher-dimensional settings. For larger datasets, the increase in TID values is more gradual, demonstrating the stabilizing effect of larger datasets.

**Training Epochs**  TID values increase with training epochs, particularly for smaller datasets. For datasets with 10,000 samples, the TID values rise steadily as the number of epochs increases, suggesting that longer training amplifies collapse errors in smaller datasets. However, for larger datasets (e.g., 40,000), the TID values plateau at a relatively low level, indicating that sufficient data can counteract the adverse effects of prolonged training.

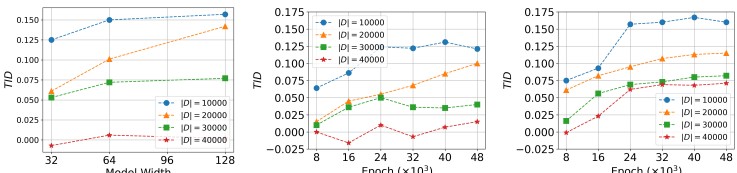

Figure 12: TIDs evaluated on ODE sampled images generated by diffusion models trained on CelebA or CIFAR10 dataset across various training settings, containing model width, dataset size, data dimension and training epoch.

## E  EXPERIMENTS ON HIGH DIMENSIONAL MoG DATASET

### E.1  EXPERIMENTAL SETTINGS

We suppose a synthetic $n$-dimension MoG dataset by:

$$\boldsymbol{x}_0 \sim 0.5 \times \mathcal{N}(\boldsymbol{x}_0| - \mathbf{1}_n, 0.2\mathbf{I}_n) + 0.5 \times \mathcal{N}(\boldsymbol{x}_0|\mathbf{1}_n, 0.2\mathbf{I}_n),$$

where $\mathbf{1}_n$ represent a vector filled with ones with a length of $n$ and $\mathbf{I}_n$ is an identity matrix with a size of $n \times n$. The dataset size are 50,000. In specific in Sec. 5.3 and Sec. 6, we choose $n$ to be 10. We use a two-layer Multilayer Perceptron (MLP) with 1000 neurons in each layer with Tanh activation. The time variable $t$ was concatenated to the data input $x$. The model was trained in stochastic gradient decent with a batch size of 2000, using the Adam (26) with a learning rate of 5e-3, and the training was conducted over 10,000 iterations.

### E.2  DERIVATION OF SCORE FUNCTION OF HIGH DIMENSIONAL MoG DATASET

We consider an MoG distribution in the following:

$$\mathbf{x}_0 \sim \frac{1}{K}\sum_{k=1}^{K}\mathcal{N}(\boldsymbol{\mu}_k, \sigma_k^2 \cdot \boldsymbol{I}),$$

where $K$ is the number of Gaussian components, $\boldsymbol{\mu}_k$ and $\sigma_k^2$ are the means and variances of the Gaussian components, respectively. Suppose the solution of the diffusin process follows:

$$\mathbf{x}_t = \alpha_t \boldsymbol{x}_0 + \sigma_t \cdot \xi \quad \text{where} \quad \xi \sim \mathcal{N}(0, \boldsymbol{I}).$$

Since $\boldsymbol{x}_0$ and $\xi$ are both sampled from Gaussian distributions, their linear combination $\boldsymbol{x}_t$ also forms a Gaussian distribution, i.e.,

$$\mathbf{x}_t \sim \frac{1}{K}\sum_{k=1}^{K}\mathcal{N}(\alpha_t\boldsymbol{\mu}_k, (\sigma_k^2\alpha_t^2 + \sigma_t^2) \cdot \boldsymbol{I}).$$

Then, we have

$$\nabla p(\boldsymbol{x}_t) = \frac{1}{K}\sum_{i=1}^{K}\nabla_{\boldsymbol{x}_t}\left[\frac{1}{2}(\frac{1}{\sqrt{2\pi}\sigma_i^2\alpha_t^2 + \sigma_t^2}) \cdot \exp(-\frac{1}{2}(\frac{\boldsymbol{x}_t - \boldsymbol{\mu}_i\alpha_t}{\sigma_i^2\alpha_t^2 + \sigma_t^2})^2)\right]$$

$$= \frac{1}{K}\sum_{i=1}^{K}p_i(\boldsymbol{x}_t) \cdot \nabla_{\boldsymbol{x}_t}\left[-\frac{1}{2}(\frac{\boldsymbol{x}_t - \boldsymbol{\mu}_i\alpha_t}{\sigma_k^2\alpha_t^2 + \sigma_t^2})^2\right]$$

$$= \frac{1}{K}\sum_{i=1}^{K}p_i(\boldsymbol{x}_t) \cdot \frac{-(\boldsymbol{x}_t - \boldsymbol{\mu}_i\alpha_t)}{\sigma_k^2\alpha_t^2 + \sigma_t^2}.$$

We can also calculate the score of $\boldsymbol{x}_t$, i.e.,

$$\nabla \log p(\boldsymbol{x}_t) = \frac{\nabla p(\boldsymbol{x}_t)}{p(\boldsymbol{x}_t)} = \frac{1/K \cdot \sum_{i=1}^{K}p_i(\boldsymbol{x}_t) \cdot \left(\frac{-(\boldsymbol{x}_t - \boldsymbol{\mu}_i\alpha_t)}{\sigma_k^2\alpha_t^2 + \sigma_t^2}\right)}{1/K \cdot \sum_{i=1}^{K}p_i(\boldsymbol{x}_t)}.$$

## F  COLLAPSE ERRORS PROPAGATION IN OTHER DATASETS

### F.1  DENSITY EVOLUTION IN OTHER DATASETS

When observing collapse errors in a distribution level, we normally need 50,000 data. Since it is expensive to sample 50,000 real image data, we hereby show the density evolution in synthetic datasets, as shown in Fig. 17.

**Experimental Settings**   The datasets in Fig. 17 contains semicircle, spiral, chessboard-shaped datasets, the synthetic MoG, and a 1D MoG datasets. The construction of semicircle, spiral, chessboard-shaped datasets, the synthetic MoG, are introduced already in Appendix. C.1. The 1D MoG datasets is the 1D version of the high-dimension MoG in Appendix. E.1 where the dimension $n$ is set to be 1. We use a three-layer Multilayer Perceptron (MLP) with 100 neurons in each layer and Tanh activation functions. The time variable $t$ was concatenated to the data input $x$. The model was trained in stochastic gradient decent with a batch size of 2000, using the Adam optimizer (26) with a learning rate of 5e-3, and the training was conducted over 10,000 iterations.

### F.2  VELOCITY MISFITTING IN OTHER DATASETS

In this subsection, we visualize the learned velocity field in both synthetic datasets and real image datasets. Here, we introduce experimental settings in this section.

**Dataset Construction**   The datasets in this subsection contain 1D-MoG dataset, MNIST, CIFAR10 and CelebA. The construction of 1D-MoG dataset follows the previous section (Appendix. F.1). To speed up the training, we set the dataset size of CIFAR10 and CelebA to be 15,000, and downsample the data to have a image size of $8 \times 8$. We also down sample the MNIST Data to be $8 \times 8$.

**Training**   We set the epoch number to be 48,000 for CelebA and CIFAR10, and 1200 for MNIST. We use Adam optimizer (26) with a learning rate of 5e-3 for CIFAR10 and CelebA, and 2e-4 for MINST. We use stochastic gradient decent with a batch size of 128 for CIFAR10 and CelebA, and 60 for MNIST.

**Model Architecture**   In our experiments, we identify error propagation in the velocity field along $t$ as a core factor contributing to collapse errors. We note that this behavior is architecture-dependent and does not occur in all model architectures. For instance, consider a 1D case $v_\theta(x, t)$; if the error propagates only along $t$, switching the variables (i.e., reordering to $v_\theta(t, x)$) could prevent error propagation along $t$. While we have not exhaustively explored all possible model architectures, we have conducted experiments on prominent architectures such as U-Net with time embeddings and MLP with time concatenation.

In this section, we use totally three model architectures for real image dataset, U-Net-raw, U-Net-reduced, and U-Net-temb. We will introduce them one by one. For CIFAR10 and CelebA experiments, the model architecture follows the default implementation in (59), but we change the channel-multiplier in U-Net to (1,2,2) instead of the default (1,2,2,2). We refer this model by U-Net-raw. We also use a simplified model in this section. The model was a two-layer U-Net, consisting of an encoder and decoder. The encoder included two convolutional layers (kernel size 3, 256 channels, Tanh activation) and a MaxPooling layer ($2 \times 2$) for downsampling. The decoder mirrored this structure with transpose convolutions for upsampling and used skip connections to combine features from the encoder. A final output convolution layers layer to reconstruct the image. To feed the time variable, we adopt two approches. In the first approach, the time variable $t$ was expanded and concatenated to the input $x$ as an additional channel. In the second approach, the $t$s are presented by a typical positional embeddings (62) implemented in (59) and then projected to embeddings by a single-layer MLP with a width of 512, then the embeddings are add to the output after each convolution. We refer these two model by U-Net-reduced-concat and U-Net-reduced-temb. For the 1D MoG dataset, we adopt the model architecture from Appendix. E.1.

To supplement the experiment on synthetic dataset, we visualize the sampling trajectories and velocity field when we set the MLP width to 10, 80, 100, 1000, respectively. We observe that when model size grows, the model misfit in high-noise regime, leading to more severe trajectory concentration, as

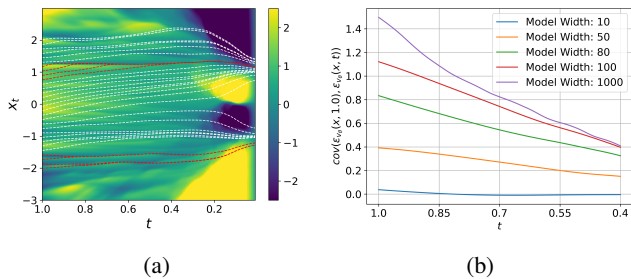

(a)                                    (b)

Figure 13: (a) Visualization of the velocity field $(v_\theta(x,t))$ when the target distribution is a 1D MoG. (b) Velocity error covariance across $t$. The covariance is calculated by comparing the error vectors of $v_\theta(x,t)$ and $v_\theta(x,1.0)$. The tested point $x$ are sampled from standard 1D Gaussian.

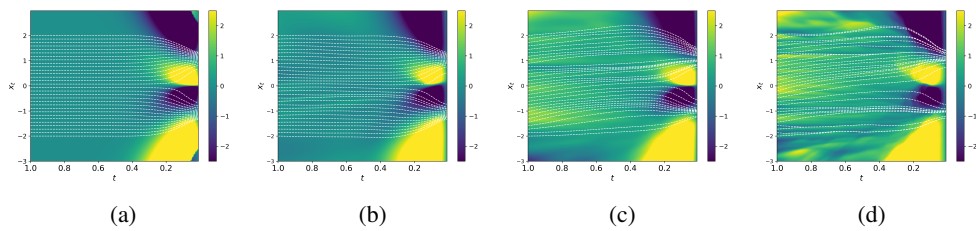

(a)                    (b)                    (c)                    (d)

Figure 14: Visualization of learned velocity fields and corresponding sampling trajectories when the target distribution is a 1d MoG. (a) Analytical solution. (b-d) learned velocity field when the three-layer Tanh MLP width is 10, 100, 1000, respectively.

shown in Fig. 13. We also calculate the error covariance to show the velocity error propagates, as shown in Fig. 14.

In Fig. 15, we show the velocity error when we train U-Net-reduced-concat on MNIST CIFAR10, and CelebA. We observed that when the model size grows, the error in high noise regime increase correspondingly. Moreover, the error propagates along $t$. When the velocity map is oscillated along $x_t$, it implies that at certain regions, the samples are directed to be closer. Once samples collapse to similar positions, it is difficult for them to escape from the collapsed region in later sampling, as the velocity field governs their dynamics identically.

In Fig. 16, we show the velocity error when we train U-Net-reduced-temb on MNIST and U-Net-raw on CIFAR10 and CelebA. Firstly, we observe that the model with various width fit score function high noise regime much better using U-Net-reduced-concat, with an absolute error around 0.01, so we cannot oberve a clear relation from the velocity error to the model size. Besides, we find when applying time embedding, the velocity error become less structured, but the error still propagate for within a short period of $t$. As we discussed, once the the data points get closer in sampling, it is hard for them to escape from similar position in later sampling. Admittedly, we also consider using time embedding to address collapse errors, but by doing some preliminary experiments, we find it less effective than the three techniques introduced in Sec.6.

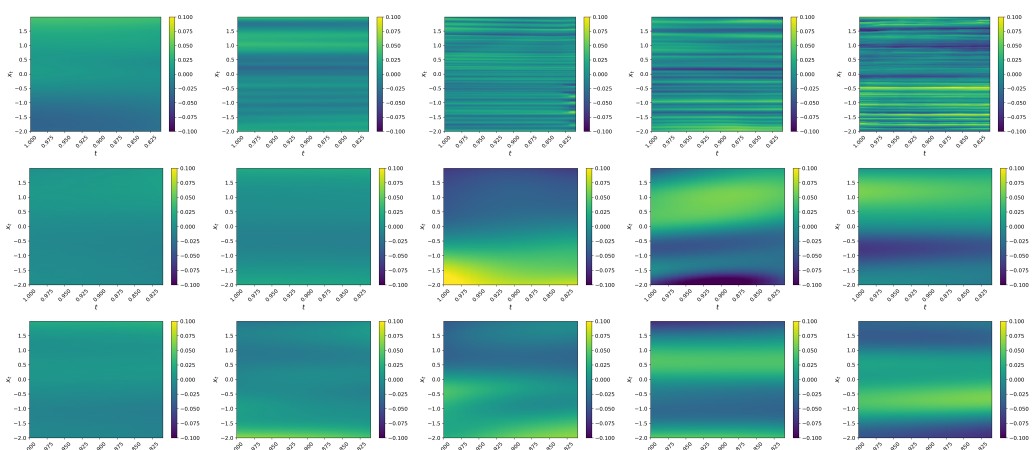

Figure 15: Visualization of the learned velocity fields $v_\theta(x_t, t)$ in high noise regimes ($t \in [0.8, 1)$) when U-Net-reduced-concat models are trained on MNIST, CIFAR10 and CelebA (top to bottom rows, respectively). The channel sizes of convolution filters in U-Net-reduced-concat models are set as 32, 128, 256, 512, and 1024, displayed from left columns to right columns.

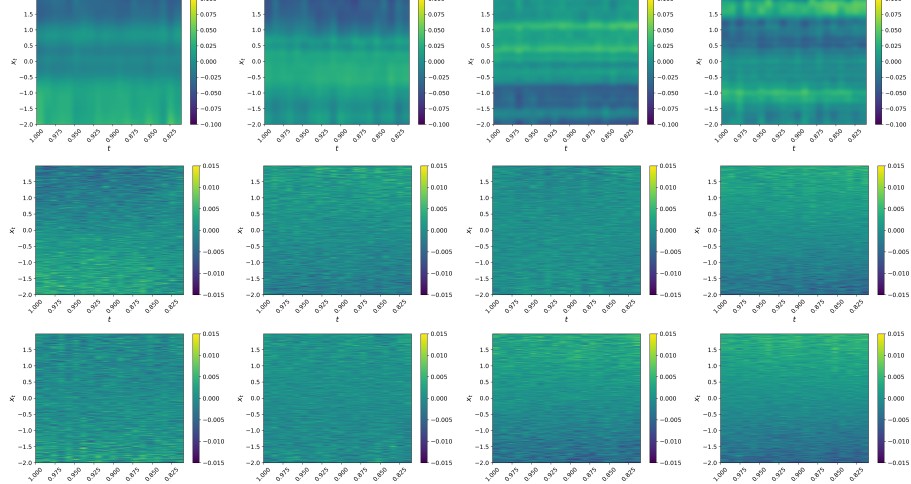

Figure 16: Visualization of the learned velocity fields $v_\theta(x_t, t)$ in high noise regimes ($t \in [0.8, 1)$) when U-Net-reduced-temb models are trained on MNIST and U-Net-raw are trained on CIFAR10 and CelebA (top to bottom rows, respectively). The channel sizes of convolution filters in U-Net-reduced-concat models are set as 32, 64, 96, 128, displayed from left columns to right columns.

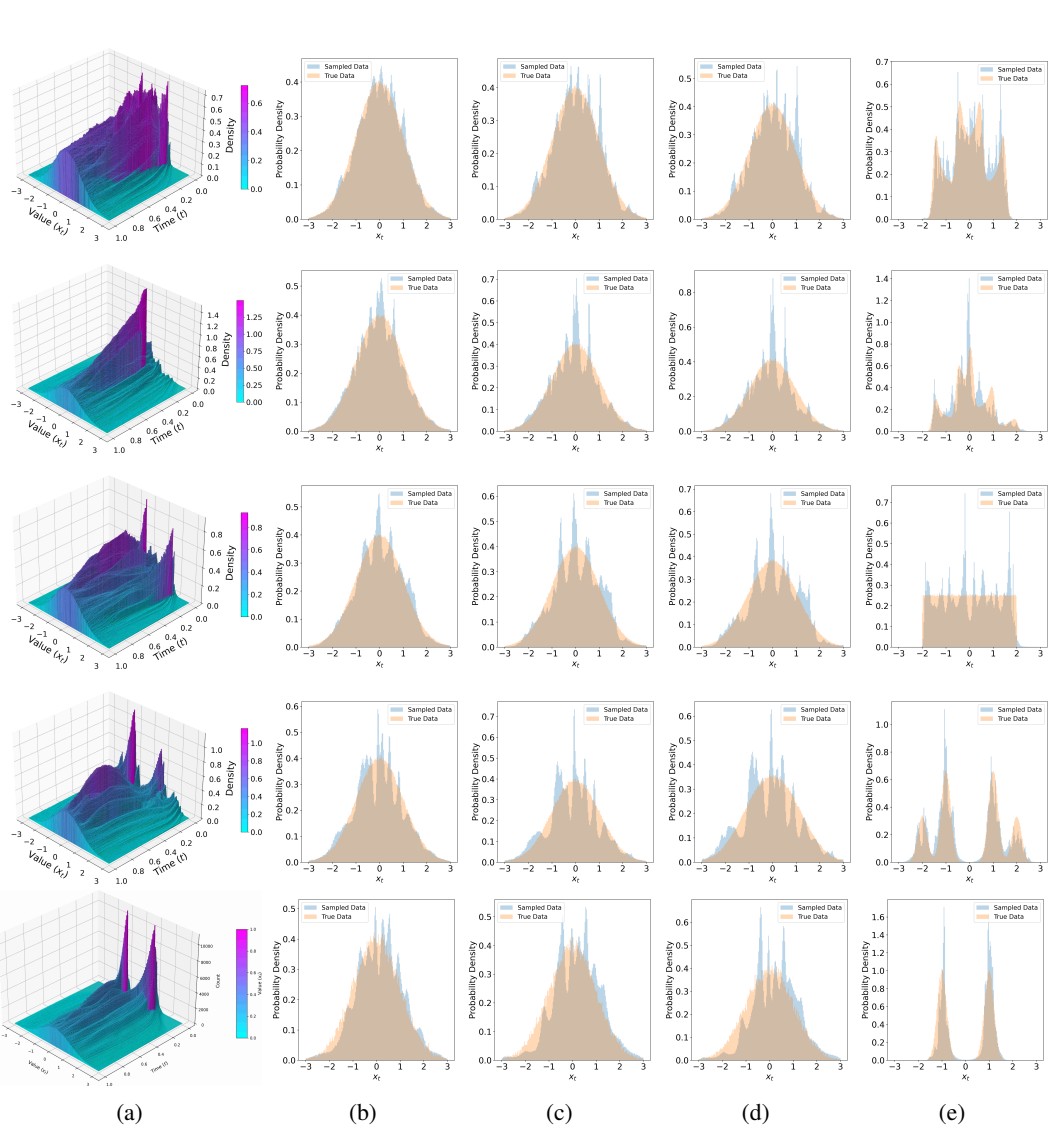

Figure 17: Visualization of the density evolution of the first dimension of data across semicircle, spiral, chessboard-shaped datasets, the synthetic MoG, and a 1D MoG datasets, during ODE-based diffusion sampling. (a) The evolution of the probability density across timesteps, starting from the Gaussian prior to the final target distribution (MoG). (b-e) Comparison of the probability density between sampled data (blue) and true data (orange) at specific timesteps $t = 0.8, 0.6, 0.4, 0.0$.

# G MORE EXPLANATIONS ON SEE-SAW PHENOMENON

## G.1 EXPERIMENTAL SETTINGS

In this section, we provide detailed descriptions of the experimental settings used in Fig. 7 and Fig. 8. Additionally, we present extended experimental results on other high-dimensional MoG datasets and different model architectures.

For the experimental settings of the synthetic high-dimensional MoG dataset, the dataset construction follows Appendix. E.1. We utilize a Multilayer Perceptron (MLP) with an equal number of neurons in each layer. The time variable $t$ is concatenated with the data input $x$. The model is trained using gradient descent with the Adam optimizer (26), a learning rate of 5e-3 , and 10,000 training iterations. To supplement our findings in Fig. 7, we conduct extensive experiments varying key parameters, including MLP width, MLP depth, MLP activation functions, MoG standard deviation, and MoG dimensionality. The velocity error is computed by averaging the Mean Absolute Error (MAE) over five independent runs. Each experimental setting is denoted in the format 'MoG Dimension-MoG Standard Deviation-MLP Depth-MLP Activation'. For example, '10-0.02-3-ReLU' refers to an experiment setting where the MoG has 10 dimensions and a standard deviation of 0.02, with an MLP comprising 3 layers and using ReLU activation.

For the experimental settings of CIFAR10 and CelebA, we follow the dataset construction and training details described in Appendix. B.1. The model architectures used follow the U-Net-reduced design, as introduced in Appendix. F.1. We report DSM losses as the average over five independent runs.

## G.2 EXPERIMENTAL RESULTS

In this section, we present our experimental results using various MLP and MoG configurations. Fig. 19 and Fig. 20 illustrate the see-saw phenomenon observed in MLPs with ReLU and Tanh activations, respectively. We observe that MLPs with ReLU activations fit the score function in high noise regimes significantly better than those with Tanh activations. Although MLPs with ReLU exhibit lower error in high noise regimes, the see-saw phenomenon persists: once the MLP starts to fit the score function better in low noise regimes, it begins to misfit in high noise regimes. The see-saw phenomenon is even more pronounced in MLPs with Tanh activations.

Furthermore, we visualize the predicted velocity function in both high and low noise regimes, confirming that the model misfits in high noise regimes by fitting the velocity function in an oscillatory manner.

To complement the results presented in Fig. 8, we also evaluate the DSM loss of diffusion models trained on the CelebA dataset, as shown in Fig. 18.

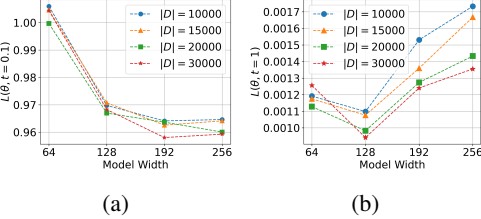

(a)  (b)

Figure 18: Diffusion loss $L(\theta, t = 0.1)$ (left) and $L(\theta, t = 1)$ (right) on CelebA with various settings on model widths and dataset sizes.

## G.3 THEORETICAL EXPLANATION ON SEESAW EFFECT

Note that for Gaussian mixture model, the ground-truth score at time $t$ satisfies (54)

$$s_t(x) = \tanh(\langle \mu_t, x \rangle)\mu_t - x, \quad \mu_t = \mu e^{-t}. \tag{1}$$

for $\|\mu\| = 1$ being the mode mean of the Gaussian mixture at time $t = 0$, $x, \mu \in \mathbb{R}^d$. Then, in order to better explain the see-saw effect, we consider the special case that $d = 1$. Besides, as it is known

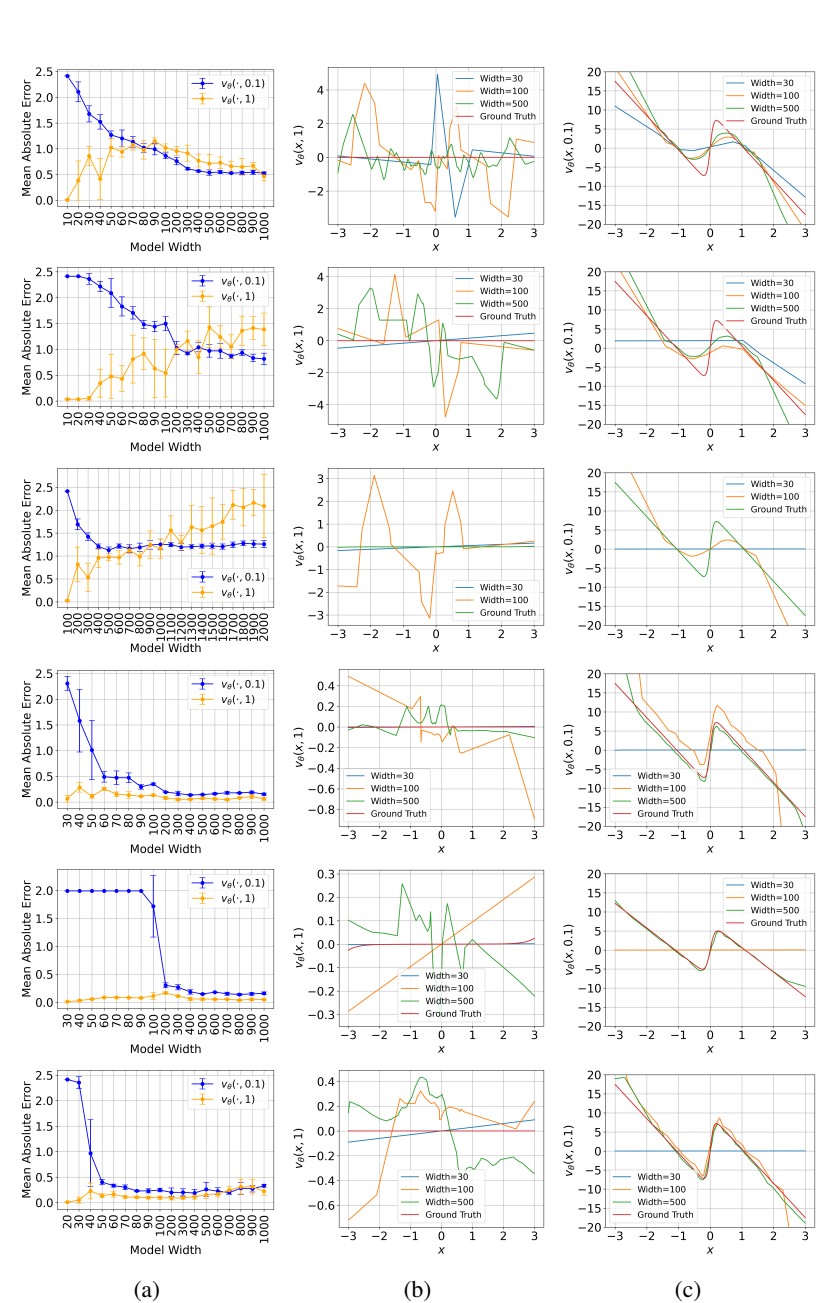

Figure 19: Visualization of velocity error when the training distribution is a high-dimensional MoG and the models are MLPs with increasing widths. For top to down the experiment settings are '5-0.02-1-ReLU', '10-0.02-1-ReLU', '25-0.02-1-ReLU', '10-0.02-2-ReLU', '25-0.2-2-ReLU', '10-0.02-3-ReLU'. The explaination of these experiments notation can be found in Appendix. G.1 (a) Mean absolute error of $v_\theta(\cdot, 1)$ and $v_\theta(\cdot, 0.1)$ along MLP widths. (c) and (d) visualize the learned $v_\theta(\cdot, 1)$ and $v_\theta(\cdot, 0.1)$, respectively, along MLP widths.

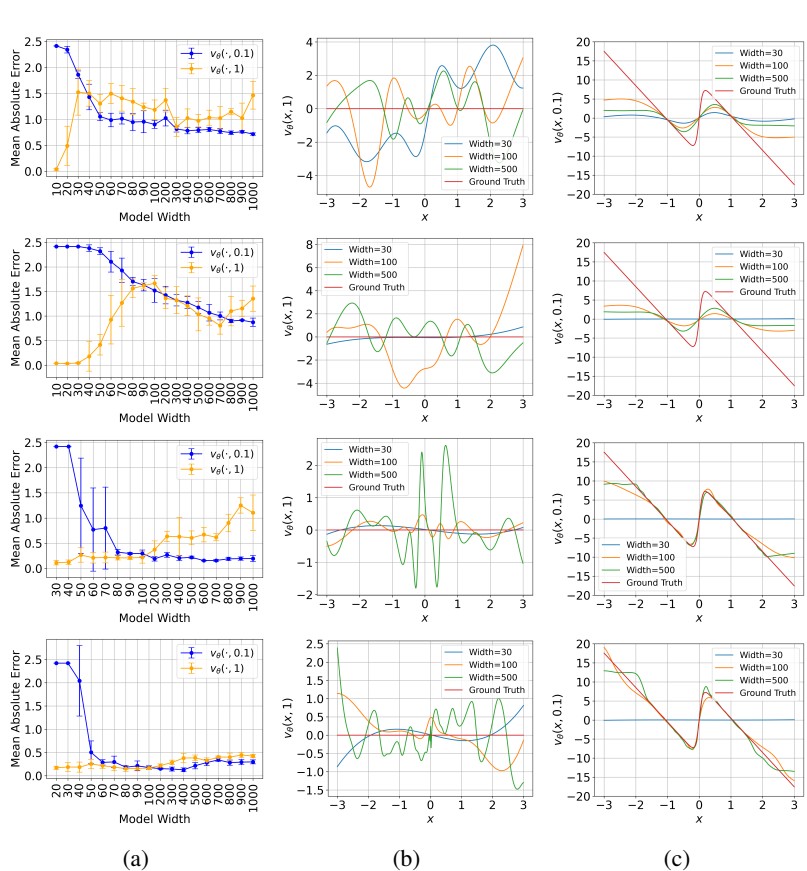

Figure 20: Visualization of velocity error when the training distribution is a high-dimensional MoG and the models are MLPs with increasing widths. For top to down the experiment settings are '5-0.02-1-Tanh', '10-0.02-1-Tanh', '10-0.002-2-Tanh', '10-0.002-3-Tanh'. (a) Mean absolute error of $v_\theta(\cdot, 1)$ and $v_\theta(\cdot, 0.1)$ along MLP widths. (c) and (d) visualize the learned $v_\theta(\cdot, 1)$ and $v_\theta(\cdot, 0.1)$, respectively, along MLP widths.

that the denoising score matching problem is equivalent to learn the optimal score function, we then resort to the following minimization problem as the score learning problem:

$$\theta^* = \arg\min_{\theta} \mathbb{E}_x\left[\|f_\theta(x, t_1) - s_{t_1}(x)\|_2^2 + \|f_\theta(x, t_2) - s_{t_2}(x)\|_2^2\right],$$

where we set $x \sim N(0, 1)$ and consider two timestamps $t_1 \to 0$ and $t_2 \to \infty$, which stand for the low-noise regime and high-noise regime, respectively. Moreover, in order to model the learner's function with different complexities (mimicking the neural network with varying neurons), we consider the following score network model:

$$f_\theta^p(x, t) = \theta_0 \cdot t + \sum_{i=1}^{p} \theta_i \cdot \mathrm{He}_i(x),$$

where $p > 0$ is used to control the complexity of the function, $\mathrm{He}_i(x)$ is the degree-$i$ Hermite polynomial, which is typically used to characterize the learnability of non-linear models under Gaussian measure. More specifically, the Hermite polynomials is a family of basis functions under Gaussian measure, i.e., it holds that

$$\mathrm{He}_0(x) = 1, \quad \mathrm{He}_1(x) = x, \quad \mathrm{He}_2(x) = \frac{1}{\sqrt{2}}(x^2 - 1), \quad \mathrm{He}_3(x) = \frac{1}{\sqrt{6}}(x^3 - 3x), \dots, \quad (2)$$

and the following orthogonality holds

$$\int \mathrm{He}_i(x)\mathrm{He}_j(x)\mu(dx) = \delta_{ij}. \quad (3)$$

Besides, it follows from Riesz-Fischer theorem (see, for example (48, Theorem 11.43)) that any square integrable function $s \in L^2(\mu)$ with respect to Gaussian measure $\mu$ can be formally expanded as

$$s(x) \sim \sum_{\ell=0}^{\infty} \alpha_\ell \mathrm{He}_\ell(x), \quad \alpha_\ell = \int s(x)\mathrm{He}_\ell(x)\mu(dx), \quad (4)$$

with $\alpha_\ell$ the $\ell^{th}$ Hermite coefficient of $s$.

Then, consider the setting that $t_1 \to 0$ and $t_2 \to \infty$, we can obtain:

- for $t_1 = 0$, we have

$$s_{t_1}(x) = \tanh(\langle \mu, x \rangle) - x = \sum_{i=1}^{\infty} \alpha_i^{(1)} \mathrm{He}_i(x),$$

  where $\alpha_1^{(1)}, \alpha_3^{(1)}, \dots, \alpha_{2k+1}^{(1)} \dots < 0$ and $\alpha_2^{(1)}, \dots, \alpha_{2k}^{(1)} \dots = 0$;
- for $t_2 \to \infty$, we have

$$s_{t_2}(x) = -x = \sum_{i=1}^{\infty} \alpha_i^{(2)} \mathrm{He}_i(x),$$

  where $\alpha_1^{(2)} = -1$ and $\alpha_2^{(2)}, \alpha_3^{(2)}, \dots, \alpha_{2k}^{(2)} \dots = 0$.

Then, based on the above results, we can further derive the optimal solutions for $\theta^*(p)$, when considering at most degree-$p$ Hermite polynomials, as follows:

$$\theta^*(p) = \frac{\theta^{1*}(p) + \theta^{2*}(p)}{2},$$

where

$$\theta^{1*}(p) = \arg\min_{\theta} \mathbb{E}_x\left[\|f_\theta^p(x, t_1) - s_{t_1}(x)\|_2^2\right], \quad \theta^{2*}(p) = \arg\min_{\theta} \mathbb{E}_x\left[\|f_\theta^p(x, t_2) - s_{t_2}(x)\|_2^2\right].$$

Then, as the Hermite polynomials are orthogonal with each other and we set $x \sim N(0, 1)$ in the above optimization problems. We can immediately obtain that $\theta_i^{1*}(p) = \alpha_i^{(1)}$ and $\theta_i^{2*}(p) = \alpha_i^{(2)}$ for all $i \leq p$.

Consequently, we can obtain the following proposition, showing the seesaw effect as $p$ increases.

**Proposition 1** *Let $\ell_1^p(\theta) = \mathbb{E}_x\left[\|f_\theta^p(x, t_1) - s_{t_1}(x)\|_2^2\right]$ and $\ell_2^p(\theta) = \mathbb{E}_x\left[\|f_\theta^p(x, t_2) - s_{t_2}(x)\|_2^2\right]$ be the score learning losses for timestamps $t_1$ and $t_2$ respectively. Then, it holds that*

$$\ell_1^p(\theta^*(p)) \leq \ell_1^{p+1}(\theta^*(p+1)), \; \ell_2^p(\theta^*(p)) \geq \ell_2^{p+1}(\theta^*(p+1)),$$

*for all $p \geq 1$.*

**Proof.** As $x \sim N(0, 1)$, based on the properties of Hermite polynomials, we can show that

$$\ell_1^p(\theta^*(p)) = \sum_{i=1}^{p} [\theta_i^*(p) - \alpha_i^{(1)}]^2 + \sum_{i>p} [\alpha_i^{(1)}]^2$$

$$\ell_2^p(\theta^*(p)) = [\theta_1^*(p) + 1]^2 + \sum_{i>1} [\theta_i^*(p)]^2.$$

Note that $\theta_i^*(p) = \frac{\alpha_i^{(1)} + \alpha_i^{(2)}}{2}$ for any $i \le p$, we can then show that

$$\ell_1^p(\theta^*(p)) = \frac{1}{4} \sum_{i=1}^{p} [\alpha_i^{(1)}]^2 + \sum_{i>p} [\alpha_i^{(1)}]^2,$$

which is strictly decreasing as $p$ increases. Besides, we also have

$$\ell_1^p(\theta^*(p)) = \frac{1}{4} [1 - \alpha_1^{(1)}]^2 + \frac{1}{4} \sum_{i>1} [\alpha_1^{(1)}]^2,$$

which is strictly increasing as $p$ increases. This completes the proof.

# H  MORE DETAILS OF POTENTIAL SOLUTIONS

## H.1  EXPERIMENTAL SETTINGS

We use the default experimental settings from Appendix B.1 and Appendix E.1, with additional necessary modifications to incorporate the three proposed techniques. Specifically, for the predictor-corrector technique, we follow (59) and apply a one-step MCMC correction after each ODE step. For the Two-Model training strategy, we duplicate the original model and set $t' = 0.6$, separating the training of high and low noise regimes. For the skip connection technique, we construct the model as follows:

$$\hat{s}_\theta(\boldsymbol{x}_t, t) = c_1 \boldsymbol{x}_t + c_2 s_\theta(\boldsymbol{x}_t, t),$$

where $c^1_{\theta_1}(t)$ and $c^2_{\theta_2}(t)$ are learnable coefficients with parameters $\theta_1$ and $\theta_2$, respectively. and $s_\theta$ is the default model introduced in Appendix B.1 and Appendix E.1 for real image and synthetic datasets, respectively.

We model $c^1_{\theta_1}(t)$ and $c^2_{\theta_2}(t)$ using a two-layer MLP with 30 neurons per layer and Tanh activation functions. Our preliminary experiments also suggest that directly setting $c^1_{\theta_1}(t)$ and $c^2_{\theta_2}(t)$ to fixed coefficients $\sigma_t$ and $1 - \sigma_t$ provides satisfactory results, where $\sigma_t$ is derived from the diffusion process solution, i.e.,

$$x_t = \alpha_t x_0 + \sigma_t \xi, \quad \xi \sim \mathcal{N}(\boldsymbol{0}, \boldsymbol{I}).$$

## H.2  VALIDATING THE CAUSE OF COLLAPSE VIA EXISTING METHODS ON REAL IMAGE DATASETS

In this section, we apply the three proposed techniques to CIFAR10 and CelebA, with necessary modifications to adapt them to real image datasets.

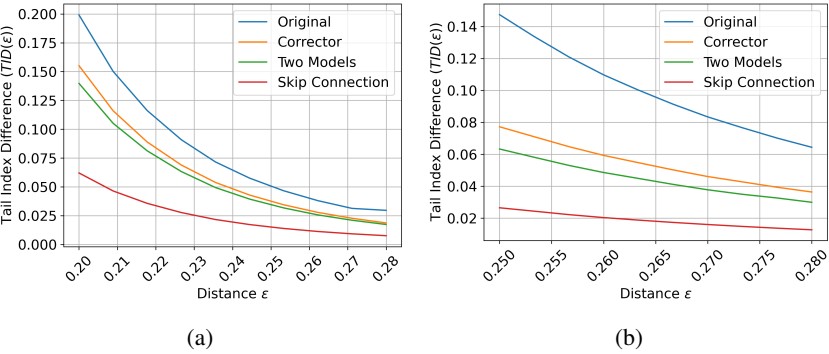

(a)                                        (b)

Figure 21: We evaluate the effectiveness of proposed techniques on mitigating collapse errors when the training datasets are CIFAR10 and CelebA. TID values evaluated across different $\epsilon$ comparing the proposed techniques (corrector, two-model training, and skip connection) and the original method. (a) and (b) shows the experimental results on CIFAR10 and CelebA, respectively.

## I  FEATURE DENSITY WHEN COLLAPSE ERRORS OCCUR

In this paper, we consistently use the $l_2$ norm as the distance metric when evaluating collapse errors. It is important to note that we choose the $l_2$ norm because we observe that collapse errors exhibit more distinctive characteristics in the raw pixel space. While many studies, particularly in image generation tasks, calculate similarity using the Fréchet distance (14) in feature space, our preliminary experiments show that collapse errors manifest differently in raw pixel space compared to feature space. In feature space, the errors appear more like a 'biased' error, which presents another interesting avenue for exploration. In this section, we show our preliminary results. In our experiments, we use InceptionV3 (61) to extract features from ODE, SDE sampled data, and training data. The features are extracted after the first Maxpooling layer, with a length of 64. Fig. 22 shows the statistics of features of selected channels. The CIFAR10 images are downsampled to $16 \times 16$ and the dataset size is set to be 10,000. We follows other experimental settings in Appendix. B.1.

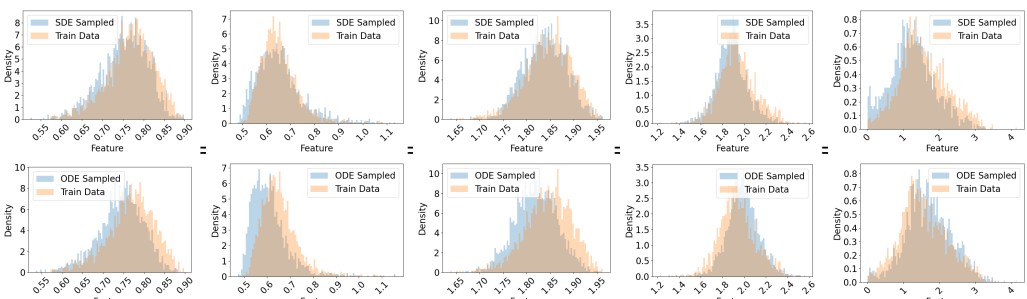

Figure 22: Visualization of InceptionV3 feature histograms of SDE sampled CIFAR10 (top row), and ODE sampled CIFAR10 (second row) using the same trained score neural network. The select feature channel indices are 1, 19, 23, 30 and 57 (from left to right)

We mention in Sec. A that a primary reason for using the $l_2$ norm as the distance metric is that diffusion sampling typically operates in the data space, making the $l_2$ norm a natural choice. However, we acknowledge that diffusion sampling can also be performed in the feature space, as in Latent Diffusion Models (46). In such cases, collapse errors may occur in the feature space, making the direct application of the $l_2$ norm in the data space less effective for evaluating them. Furthermore, this raises two important questions: If collapse errors arise in the feature space, (1) how do they behave in the data space? (2) can commonly used metrics on data space such as FID (18) and IS (52) effectively capture them? We leave these as open questions for future investigation.

## J  COLLAPSE ERRORS ON OTHER DIFFUSION PROCESSES

In this paper, we primarily conduct experiments on the typical VP diffusion process to maintain variable control. We also observe collapse errors in other diffusion processes, including the linear (35; 34) and Sub-VP (59) diffusion processes. Fig. 23 illustrates the occurrence of collapse errors on the high-dimensional MoG dataset. We follow all experimental settings detailed in Appendix E.1, except for the choice of the diffusion process.

## K  COLLAPSE ERRORS ON OTHER DETERMINISTIC SAMPLERS

To demonstrate collapse errors, our study mainly study a specific deterministic sampler: the reverse ODE sampler. It is important to note that while we focus on this sampler for controlled variable analysis, this does not imply that collapse errors are absent in other deterministic samplers. In this section, we also present evidence of collapse errors when using DDIM (57) and the second-order DPM solver (37). Our experiments do not reveal significant differences in collapse errors among these samplers, though exploring how specific deterministic samplers influence collapse errors remains an interesting direction for future research.

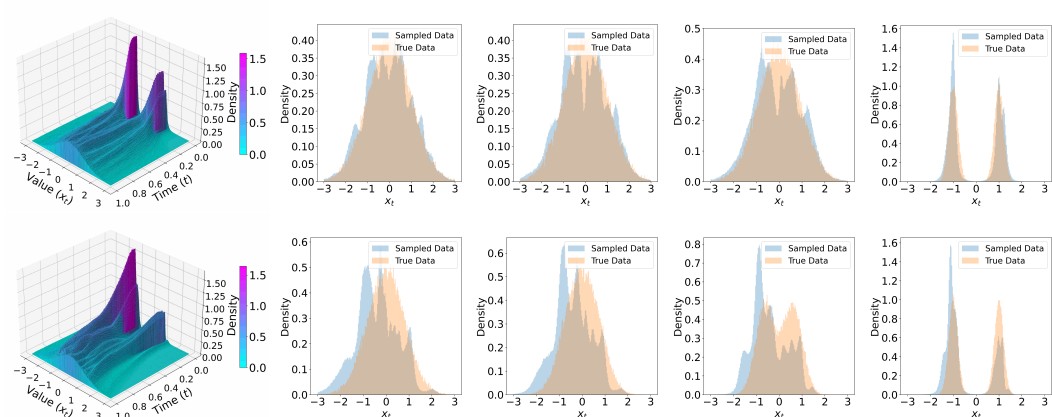

Figure 23: Visualization of the density evolution of the first dimension of data on the synthetic MoG datasets, during ODE-based diffusion sampling. The diffusion processes are Sub-VP (top row) and linear (lower row). (a) The evolution of the probability density across timesteps, starting from the Gaussian prior to the final target distribution (MoG). (b-e) Comparison of the probability density between sampled data (blue) and true data (orange) at specific timesteps $t = 0.8, 0.6, 0.4, 0.0$.

Figure 24 illustrates the occurrence of collapse errors on the high-dimensional MoG dataset. We follow all experimental settings detailed in Appendix E.1, with the only difference being the choice of samplers.

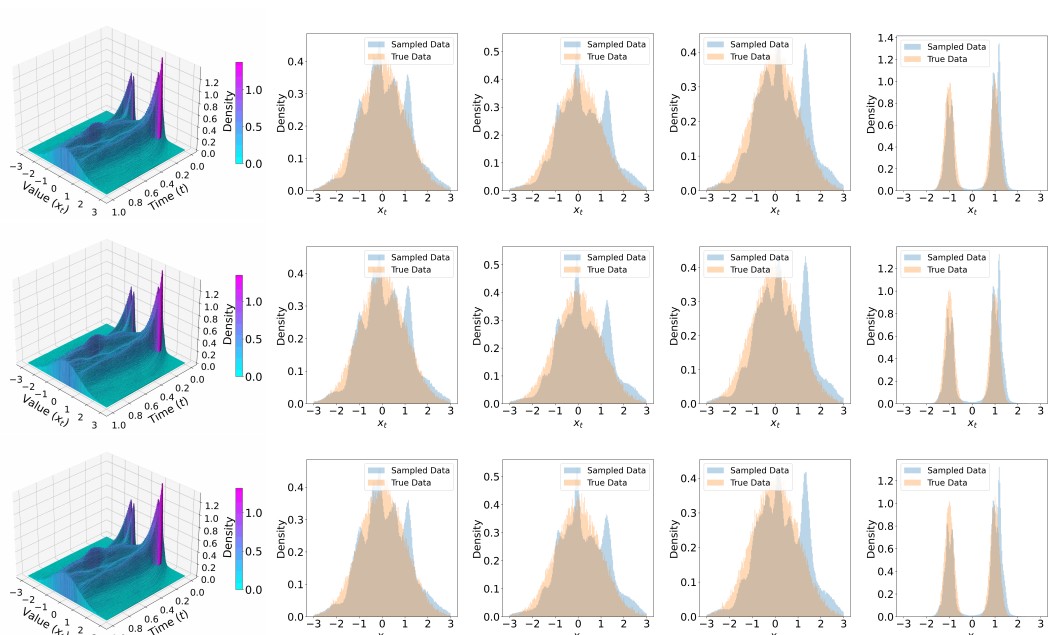

Figure 24: Visualization of the density evolution of the first dimension of data on the synthetic MoG datasets, using different deterministic samplers. The deterministic samplers are ODE (top row), DDIM (middle row) and second-order DPM (lower row). (a) The evolution of the probability density across timesteps, starting from the Gaussian prior to the final target distribution (MoG). (b-e) Comparison of the probability density between sampled data (blue) and true data (orange) at specific timesteps $t = 0.8, 0.6, 0.4, 0.0$.

## L    DISCUSSION ON THE TID METRIC

Our TID evaluation involves pairwise distance calculations, which may raise concerns regarding computational cost. However, to compute TID, we only need to calculate pairwise distances within a subset of the dataset. We find that using a subset of 2,000 samples is sufficient to reliably demonstrate collapse errors. Moreover, as shown in Sec. 5.3, collapse errors can be identified through the density distribution of a single dimension of the dataset. This suggests that evaluating collapse errors does not necessarily require full pairwise distance calculations across all dimensions; instead, computing distances within a single dimension may be sufficient. Finally, we do not expect computational cost to be a significant concern for TID estimation, as it can be efficiently computed using a reduced dataset subset and a single dimension of the data.

## M    USE OF LARGE LANGUAGE MODELS

Large language models (LLMs) were used only to refine the writing and improve clarity of presentation. They did not contribute to research ideas, methodology, implementation, or experimental analysis.

