# OpenReview forum: "On the Collapse Errors Induced by the Deterministic Sampler for Diffusion Models"
_ICLR.cc/2026/Conference — ICLR 2026 Conference Withdrawn Submission_

### Official Review · Reviewer_cgPk · 2025-10-28

**Soundness:** 3
**Presentation:** 3
**Contribution:** 3
**Rating:** 6
**Confidence:** 3

**Summary:**

In this article, the authors study the collapse error in diffusion models, a phenomenon where deterministic samplers excessively concentrate probability mass in the data space. A specific metric, the Tail Index Difference (TID), is proposed to evaluate this collapse error. TID can be efficiently computed and aligns well with the FID metric. Using TID, the authors demonstrate the qualitative and quantitative patterns of collapse error and attribute it to the interplay between deterministic samplers and score misfitting. Several techniques are proposed to mitigate the collapse error, and their effectiveness validates the article's reasoning for its occurrence.

**Strengths:**

- **Originality**: A new index that evaluates the collapse error of diffusion models is proposed; causes of collapse error are investigated.
- **Quality**: Extensive numerical experiments are conducted to verify the reasoning of the causes of collapse error; the related work is comprehensive.
- **Clarity**: The motivation for the new index is clearly articulated, and the analysis of the causes of collapse error is convincingly verified by the experimental results.
- **Significance**: Collapse error is a critical issue of diffusion models.

**Weaknesses:**

- The investigation of the collapse error of diffusion models is only based on empirical experiments, while the theoretical understanding is limited.
- The techniques for mitigating collapse error are existing methods, which indicates the contribution in terms of algorithms is limited.

**Questions:**

- Please provide the reference for the existing "two model" technique.

---

### Official Review · Reviewer_bc29 · 2025-10-31

**Soundness:** 2
**Presentation:** 3
**Contribution:** 2
**Rating:** 2
**Confidence:** 4

**Summary:**

This paper studies the "collapse" behaviors of the determinstic sampler in diffusion models sampling. The author investigated the underlying reasons that caused this collapse, which is an interplay between imperfect score learning and deterministic sampling mechanism. Another metric is proposed to evaluate the generation quality which is theorectically grounded and performs similarly as FID. The author validates their hypothesis through diferent sets of experiments, following the score network and determinstic sampling setup in Song-SDE 2021.

**Strengths:**

The "collapse" behaviors of the determinstic sampler in diffusion models sampling is indeed important and worth studying. This paper did a good work in investigating the cause of error following Song-SDE 2021 setup of determinstic sampler and score network training, which is a representative baseline to start from. The investigation on the underlying reasons about the score imperfect training is also interesting. The experiments on the Song-SDE 2021 setup is pretty comprehensive.

**Weaknesses:**

First, the studied problem is not new and has been widely noticed by the iterature, e.g. in Song SDE. I am especially puzzled about the "collapse" behaviour naming. Usually generation collapse refers to a lack of diversity instead of a large FID/TID score. Here in the paper, the argument is essentially larger FID under determinstic sampler compared to stochastic sampler with the same score network, which has been clearly claimed already by the Song's SDE paper. The paper's contribution lies to investigate the possible reasons for such parameterization, which makes the overall contribution not quite significant.

Second, a big concern for this paper is that it only studied a important but too outdated baseline setup, whose findings are suspectible to be applicable to modern diffusion models designs. For example, EDM (Karras 2022) already did several improvements in the chosen deterministic sampler and score network parameterization, and suggests that determinstic samplers can actually be better than stochastic samplers in terms of FIDs. This is contradictory to this work's findings, which suggests that the paper's findings may be only applicable to the Song SDE setup. More experiments on EDM based parameterization and even flow matching is needed to showcase if modern design space of diffusion models still suffer from such collapse.

Third, though mentioned already in the paper, there are no suggested methods to improve the performance of the determinstic samplers. Especially better noise schedules and score model architecture with skip connections have already been proposed in EDM, which makes methods in Section 6 a replication even without experimental validation.

**Questions:**

1. Why do we need TID if the empirical performance is similar to the performance of FID?
2. What is the quantitative results under the experiments setups using more advanced determinstic samplers beyond Euler sampler?
3. What will be the experimental results for modern diffusion models like EDM based parameterization ord Flow Matching?

---

### Official Review · Reviewer_JYpP · 2025-10-31

**Soundness:** 3
**Presentation:** 2
**Contribution:** 2
**Rating:** 2
**Confidence:** 3

**Summary:**

This paper studies the "collapse errors" in diffusion models, referring to cases where the sampled data is overly concentrated locally. The authors argue that such behaviors arise when score fitting in the low noise regime adversely impacts that in the high noise regimes (dubbed the "see-saw" effect). The authors further define the Tail Index Difference (TID) metric to quantify the collapse errors in experiments on synthetic and real image datasets, and then show that they can be reduced through techniques including replacing deterministic samplers with stochastic ones, having separate score models for low and high noise regimes, as well as adding residual connections to the score models.

**Strengths:**

The formalization of the collapse error through the TID metric is novel to my knowledge.

**Weaknesses:**

1. The collapse error boils down to an underfitting behavior in the training of the neural network score estimator. The authors named it the see-saw effect to emphasize the tradeoff between the low- and high-noise regimes, but all together it reflects the inability of the NN to fully minimize the score matching loss that is aggregated across the different noise levels. Meanwhile, as the authors demonstrate, this issue can be reasonably mitigated by relatively simple techniques such as adding skip connections to the NN, which have already been adopted quite commonly (in fact, there is a variety of architectural choices that can be made to improve the training of diffusion models, see ref. 24). As such, I am unsure about the relevance of this work to the current practice of diffusion models.

2. Because the phenomenon under investigation is related to underfitting in NN training, it will be relevant to also include results on how the phenomenon depends on the number of training epochs.

**Questions:**

1. Figure 8 (b) shows that the score learning loss is higher when the model width is larger, which seems surprising since we generally expect the expressivity of the neural network to improve as its width increases. What could be the explanation?

2. The presentation of the experiment results also needs to be improved. For instance, in Figure 3 (a), because of the overlaps, it is unclear whether the blue bars always no taller than the green ones, or perhaps they are actually taller when x=0 or 1.

---

### Official Review · Reviewer_swB7 · 2025-10-31

**Soundness:** 3
**Presentation:** 3
**Contribution:** 3
**Rating:** 6
**Confidence:** 4

**Summary:**

This paper diagnoses a failure mode of deterministic (ODE) sampling in diffusion models that the authors call collapse error. collapse error is the tendency of ODE samples to concentrate in small regions of data space relative to the training distribution or SDE samples. The paper documents the phenomenon qualitatively and quantitatively. A new Tail Index Difference (TID) metric based on Hill’s estimator is introduced. The collapse error is shown to start in low-noise regime but propagates with time. Error collapse is linked to the interplay between ODE dynamics and score misfitting in the high‑noise regime caused by a training “see‑saw” effect. A simple theoretical proposition on the see‑saw effect (for a GMM setting) supports the story.

**Strengths:**

- The paper isolates a practical failure mode in widely‑used deterministic samplers, whose popularity stems from speed, controllability, and trajectory “straightness.” Studying their limitations is valuable for real systems and for methods that rely on determinism to learn consistency between long/short trajectories (e.g. consistency and shortcut models). The background clearly motivates why ODE paths are important.
- Collapse is shown at the sample level and at the distribution level on synthetic data. This makes the phenomenon intuitive and reproducible.
- The TID metric formalizes “local over‑concentration” and trends track FID as a secondary check which increases credibility.
- See‑saw effect is interesting. As model capacity grows, low‑noise fits improve while high‑noise fits worsen, even though the latter task is nearly an identity mapping under epsilon‑prediction.
- The exposition is structured and figures are informative.

**Weaknesses:**

- The paper explicitly does not propose a new sampler or training objective aimed at eliminating collapse, but rather uses existing techniques to validate the cause.
- The formal result Proposition 1. supports the see‑saw effect in a specific simple setting; it is not yet a general theorem that deterministic sampling must collapse under realistic assumptions. Theoretical guarantees (or impossibility results) would strengthen the paper.
- The ODE is run with 100 steps versus 1000 steps for SDE (Euler) in the main image experiments (App. B.1). It would help to weep ODE/SDE step budgets and include more modern ODE solvers (high‑order, perhaps) at comparable compute.
- TID depends on neighbor counts within radius ε and on the top‑k order statistics in Hill’s estimator; the choice of epsilon and k (or robustness to them) could be more documented.

**Questions:**

- Have you tried targets defined up to a normalizing constant? ? One might expect similar ODE over‑concentration without SDE noise. If not tested, what’s your expectation?
- To what extent does the ODE/SDE divergence mirror known differences between deterministic vs. stochastic training regimes (e.g., rectified flows vs. diffusion, or NF/flow vs. diffusion when sampling unnormalized targets)? Any insight from your see‑saw lens?
- In Fig. 2 and Fig. 11, high‑density regions appear connected by trajectories through low‑density areas (like filaments?). Is this a direct consequence of the velocity‑error you measure? Could you comment on this?

---

### Note · Authors · 2025-12-02

I have read and agree with the venue's withdrawal policy on behalf of myself and my co-authors.